# Code Editing from Few Exemplars by Adaptive Multi-Extent Composition

## Abstract

This paper considers the computer source code editing with few exemplars. The editing exemplar, containing the original and modified support code snippets, showcases a certain editorial pattern, and code editing adapts the common pattern derived from few support exemplars to a query code snippet. In this work, we propose a novel deep learning approach to solve this code editing problem automatically. Our learning approach combines edit representations extracted from support exemplars and compositionally generalizes them to the query code snippet editing via multi-extent similarities ensemble. Specifically, we parse the support and query code snippets using language-specific grammar into abstract syntax trees. We apply the similarities measurement in multiple extents from individual nodes to collective tree representations for query and support sample matching, and ensemble the matching results through a similarity-ranking error estimator. We evaluate the proposed method on C# and Python datasets, and show up to 8.6% absolute accuracy improvements compared to non-composition baselines.

## 1 Introduction

In recent years, a surge of interest has been witnessed in applying machine learning techniques to code editing (Zhao et al., 2019; Chen et al., 2019; Dinella et al., 2020; Chakraborty et al., 2020; Yasunaga & Liang, 2020). Code editing in software engineering intends to revise the design, structure, functionality, or implementation of existing programming codes into a desirable pattern. To maintain high-quality code projects or merge several repositories, the programmers typically fix one or few code snippets, and expect the same revision automatically applied to other places in demand over the whole project. Automating this process can facilitate a broad range of programming applications such as code migration, refactoring, version update, and bug repair.

The problem of code editing with exemplar(s) aims to adopt the common editorial pattern from given exemplar(s) to a query code snippet. The editorial pattern of an exemplar describes the type of change between two code snippets, and implies the underlying intent of making a kind of edit for a specific purpose, e.g., updating some keywords or reformatting redundant brackets. Pink rectangles in Figure 1 provide an example of editorial patterns with the comparisons between previous and edited code snippets. We also present a list of editorial patterns that appear in our dataset in Appendix A for reader understanding. This problem is conceptually similar to programming by example in software engineering (Menon et al., 2013; Meng et al., 2013), and recently has been studied by Yin et al. (2019); Brody et al. (2020); Yao et al. (2021) in a deep learning fashion. Technically, their methods parse the source code as an abstract syntax tree using language-specific grammar, apply one-time or sequential edit actions over the tree, and always keep the modified code snippet satisfying grammatical rules. In particular, all these works focus on performing code editing with exact one editing exemplar, i.e., the *one-shot* setting.

While the above studies have demonstrated promising results, simply adapting the editorial pattern from one single exemplar can lead to poor generalization or even incorrect code editing. Figure 1 presents two cases. In Case (a), when a model observes only the Support #1 exemplar, it may intuitively interpret the code refactoring rule as replacing the conditional expression ("`if (s==null)...`") by a coalesce one ("`...??...;`") only for a newly-declared variable ("`string s=...`"). Consequently, applying the model to a query input could mistakenly transform the newly-declared variable ("`string title=...`") rather than its re-assigned correspondence

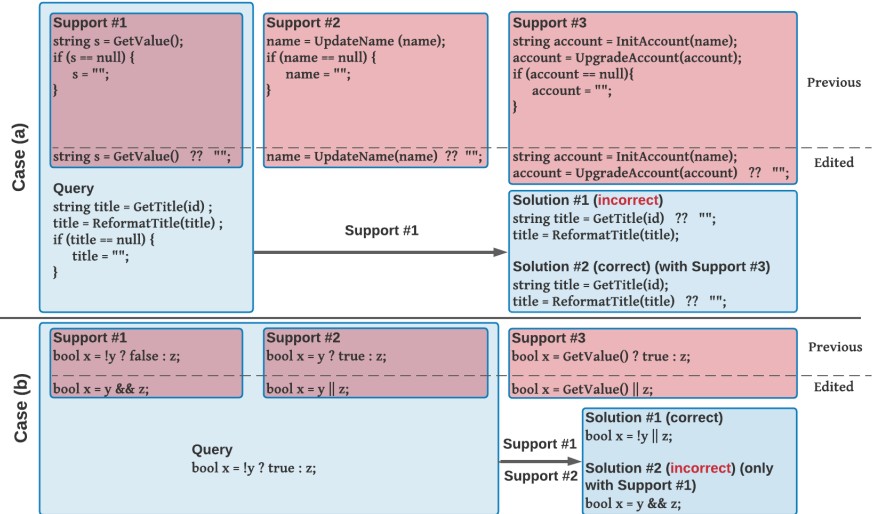

Figure 1: Two illustrative cases of our motivation for code editing learning with few exemplars, rather than with only one. The purple rectangles on the right part show the incorrect and correct solutions without or with multiple exemplars composition, respectively, in both Case (a) and (b). The editing with one exemplar may be dominated by the inductive bias learned from limited samples and thus lead to an infeasible solution. With few support exemplars, the desirable solution can be reached.

("`title=ReformatTitle(...)`"; see Solution #1). However, Case (b) demonstrates a situation where two editing exemplars turn to be sufficient for an editing model to capture the logical transformation over code tokens. The foregoing articulates the necessity of leveraging few but not one exemplar to adapt an editorial pattern to a new code snippet. The *few* exemplars help the editing escape from the inductive bias learned from one exemplar, and enhance the generalized capacity for other snippets with various coding contexts.

In light of this, in this paper, we consider the computer source code editing with few exemplars. The editing of a query code snippet is triggered by few editing exemplars that containing the original and modified support code snippets. Different from the above one-shot code editing scenario, few exemplars, even within the same editorial pattern, perform divergent coding contexts and edit actions. Therefore, it is not eligible to adapt the edit actions directly from an arbitrary exemplar. For this scenario, the major challenge lies in *how to identify and match helpful support exemplars for a query input and compose them to guide the editing*.

To address this challenge, we propose a novel approach that combines edit representations extracted from exemplars and compositionally generalizes them to the query code snippet editing via multi-extent similarities ensemble. Specifically, we parse the support/query code snippets using language-specific grammar into abstract syntax trees, where editing actions are executed on nodes. Based on this, we model the similarity between the representations of the support and query abstract syntax trees. We notice that among few exemplars, some editing action only happens in a local region, while concurrently some editing action relies on a general coding context. This inspires us to consider multi-extent similarities between the representations of the support and query abstract syntax trees. We design a $\lambda$-softmax function by scaling the importance of nodes in an abstract syntax tree, i.e., $\lambda \to 0$ means the support/query code snippet is denoted by tree representation, while $\lambda \to \infty$ means the support/query code snippet is denoted by one single node representation in the abstract syntax tree. By controlling $\lambda$ the intermediate positions between individual nodes and a collective tree, we provide multiple intermediate representational query-support matchings, then adaptively ensemble outcomes from these extents through a similarity-ranking error estimator for a robust composition. We use the term *multi-extent* to describe the scope of query-support matching at various levels from nodes to a collective tree representation.

We evaluate the algorithmic performance on two code editing datasets, one in C# (Yin et al., 2019) and one in Python (collected by us). We implement the proposed multi-extent exemplar composition mechanism on top of the state-of-the-art Graph2Edit model (Yao et al., 2021). On both datasets, our

model outperforms baseline methods by 8.0-10.9% in terms of absolute accuracy. In addition, our experimental results show that (1) enabling few support exemplars can dramatically improve code editing; (2) compared with treating all support exemplars equally, precisely capturing the code snippet similarity and compositing support exemplars are crucial for code editing; and (3) the multi-extent approach offers better performance compared to its single-extent counterpart and other existing methods. Our collected *PyFixer* dataset and model implementation will be released.

## 2 PRELIMINARY

In what follows we shall use $C_-$ and $C_+$ to represent the abstract syntax trees of the input (previous) and the output (edited) code snippets. We address the problem of code editing learning with few exemplars. Giving a set of $K$ support exemplars with an identical editorial pattern $\{(C_-^{s_k}, C_+^{s_k})\}_{k=1}^K$, the goal is to adapt the editing from support set to a query snippet $C_-^q$ and obtain its desirable $C_+^q$.

Here we introduce one of the state-of-the-arts edit learning models, Graph2Edit (Yao et al., 2021), that builds the one-shot code editing framework with three major components. (1) A tree encoder based on Gated Graph Neural Networks (Li et al., 2015) to embed the input abstract syntax tree $C_-$ into node representations $Z \in \mathbb{R}^{N \times D}$, where $N$ is the number of nodes in the tree and $D$ is the latent feature dimension. The whole tree embedding $t$ can be obtained by a graph pooling operation over all node representations; (2) An editing encoder to embed input $C_-$ and corresponding output $C_+$ into a vector $f_\Delta(C_-, C_+) \in \mathbb{R}^D$ to represent the entire $L$ sequential editing actions (four types of actions: add/delete a node, add a subtree, or stop editing) from $C_-$ to $C_+$ as well as the procedural editorial pattern; (3) A decoder for editing action prediction working conditionally on the tree embedding $t$ and editing representation $f_\Delta(C_-, C_+)$. Predictions include the operational type, the executive location on abstract syntax tree, and associated action values, e.g., to be replaced by which node. Denote $t_{1:l} = (t_1, \ldots, t_l)$ as the tree embedding history along the sequential editing, and $a_{1:l} = (a_1, \ldots, a_l)$ as the editing action history until step $l$. Formally, the whole predictions of actions can be represented as a conditional likelihood $\Pr(a_{1:L}|f_\Delta(C_-, C_+), t_1) = \prod_{l=1}^L \Pr(a_l|f_\Delta(C_-, C_+), t_{1:l})$. An earlier model Graph2Tree (Yin et al., 2019), where Graph2Edit derives from, shares the same methodological concept but edits the input in one-pass, instead of sequentially predict and apply the edit actions.

The Graph2Edit framework is trained in a self-reconstruction paradigm. The training feeds the tree encoder and the editing encoder with a pair of $C_-$ and $C_+$, and maximizes the likelihood of the whole predictions of actions towards the shortest sequential actions produced by dynamic programming. The training details with the dynamic programming algorithm can be found in Yao et al. (2021). The training only sends the vanilla editing encoder with the ground-truth from one exemplar, while $\{(C_-^{s_k}, C_+^{s_k})\}_{k=1}^K$ with an identical editorial pattern are hard to collect from a wild code dataset.

Based on the training, Graph2Edit is mainly designed for code editing with only one support exemplar. It is struggled to directly tackle few supports since few exemplars, even within the same editorial pattern, perform divergent coding context and edit actions. Unfortunately, we argue the one-exemplar case lacks the capacity of compositional generalization on editorial patterns as we discussed within the introductory part. We also empirically evidence this claim in our experimental section. Our method composites $f_\Delta(C_-^{s_k}, C_+^{s_k})$ from support set based on a multi-extent query-support matching to form the query edit representation $f_\Delta^q$ for decoding purpose which done by the code editing frameworks. In the next section, we elaborate on our proposed adaptive multi-extent similarity ensemble on exemplar composition to address the foregoing challenges in computer code editing with few exemplars.

## 3 ADAPTIVE MULTI-EXTENT COMPOSITION

We assume that if the input tree of a support exemplar enjoys more similarity with the query input tree, the query snippet is more likely to adopt the support exemplar's edit representation towards correct code editing. Based on this, we address compositional learning from a query-support matching perspective. Deriving from the basis of code editing reference, some editing can be adapted depending on individual node representation and regardless of its neighbor nodes and context. For instance, the removal of a redundant bracket should partially be independent of the inside context, or the change of some outdated function calling mode in a previous language version should be invariant to the host object. On the other hand, some editing highly relies on the contextual information

among several objects and connections therein, leading the editing learning to be determined by a collective representation. Continuing with this analysis, we design a multi-extent measurement to perform multiple intermediate query-support matching between individual nodes and the collective tree. Specifically, we measure the multi-extent similarities between query and support snippets in the tree embedding space, where we can define the coverage of nodes in a tree we want to involve for matching (from a single node to every node), then ensemble and learn a convex combination of the editing representations from the support set to maximize the adaptability for query snippet editing. In the coming subsections, we introduce the query-support matching over abstract syntax tree, extend it with multi-extent representation in an ensemble mechanism, then illustrate the meta-learning paradigm and inference procedure.

### 3.1 QUERY-SUPPORT MATCHING: FROM COLLECTIVE TREE TO INDIVIDUAL NODE

Instead of treating each node equally, our model softly emphasizes some nodes in the query snippet when meet a similar node in a support exemplar, and vice versa. Considering one query snippet and a set with $K$ support exemplars, let $z_n^q$ and $z_n^{s_k}$ denote the $n$-th node representation in the abstract syntax tree of the query and $k$-th support snippet, respectively. Let $\varphi_\theta(\cdot, \cdot) : \mathcal{Z} \times \mathcal{Z} \to \mathbb{R}$ be a query-support node matching similarity function (e.g., a neural network) with a learnable parameter $\theta$ that takes the node representation from both sides as input. Note that $\varphi_\theta(\cdot, \cdot)$ does not have to be non-negative or symmetrical since the measurement across query and support can be deemed as directional. We compute the initial query-support node matching activation $m_n^{q s_k}$ and $m_n^{s_k}$ as follows:

$$m_n^{q,s_k} := \max \left\{ \varphi_\theta \left( z_n^q, z_i^{s_k} \right) \right\}_{i=1}^{N_s} \text{ and } m_n^{s_k,q} := \max \left\{ \varphi_\theta \left( z_i^q, z_n^{s_k} \right) \right\}_{i=1}^{N_q}, \tag{1}$$

where $N_q$ and $N_s$ are the numbers of nodes in the query and support abstract syntax trees. The maximum operation over the set of one-to-all node emphasizes the matching between individual nodes, and returns a high activation when there is at least one good match in the counterpart. The activation represents the likelihood that a query node appears in a support snippet and vice versa.

To capture the multi-extent matching, we design a $\lambda$-softmax function by scaling the importance of nodes in an abstract syntax tree, and seek the query-support matching at the intermediate position between individual nodes and a collective tree:

$$\sigma_{n,\lambda}^{q,s_k} := \frac{\exp\left(\lambda m_n^{q,s_k}\right)}{\sum_{i=1}^{N_q} \exp\left(\lambda m_i^{q,s_k}\right)} \text{ and } \sigma_{n,\lambda}^{s_k,q} := \frac{\exp\left(\lambda m_n^{s_k,q}\right)}{\sum_{i=1}^{N_{s_k}} \exp\left(\lambda m_i^{s_k,q}\right)}. \tag{2}$$

$\sigma_{n,\lambda}^{q,s_k}$ denotes the parameterized node activation from the query snippet to the $k$-th support snippet, and a similar expression can denote as $\sigma_{n,\lambda}^{s_k,q}$. Based on the above normalized activation for individual node representation, the collective tree representation of the query and support snippets can be calculated using a weighted average pooling as follows:

$$t_\lambda^q := \frac{1}{K} \sum_{n=1}^{N_q} \sum_{k=1}^{K} \sigma_{n,\lambda}^{q,s_k} z_n^q \text{ and } t_\lambda^{s_k} := \sum_{n=1}^{N_{s_k}} \sigma_{n,\lambda}^{s_k,q} z_n^{s_k}. \tag{3}$$

The representation of the query snippet has the summation of matching values on nodes from all support exemplars. Combining Eq. (2) and (3), an intuitive interpretation on variable $\lambda$ raises: a larger $\lambda$ denotes greater domination of matched individual nodes in the final tree representation, i.e., the sharpness of the outcomes after $\lambda$-softmax normalization. $\lambda \to 0$ preserves the final query/support representation as to their initial tree representation, meaning it calculates a more smoothly weighted average of node representations over the entire tree. $\lambda \to \infty$ represents the final query/support representation approximately with only one single node representation at most of the time, where the node is selected upon the maximum activation from Eq. (1). Setting a large $\lambda$ value implies a smaller coverage over the tree. This coverage scaling property holds due to the monotonous increase of the first-order gradient of the exponential function. We will tackle the $\lambda$ value selection problem in the next subsection. Currently, we reach to the generation of edit representation for query sample editing, and the updated tree representation yields the expression:

$$f_\Delta^q := \sum_{k=1}^{K} \phi_\theta(t_\lambda^q, t_\lambda^{s_k}) f_\Delta(C_-^{s_k}, C_+^{s_k}), \tag{4}$$

where $\phi_\theta(\cdot, \cdot)$ is a similarity measure over query and support tree representations and should satisfy the convexity in the above combination. Note that for a simplified expression, we do not distinguish the learnable parameter $\theta$ for each module but their parameters are isolated.

## 3.2 MULTI-EXTENT COMPOSITION AND ENSEMBLE

The $\lambda$-softmax in Eq. (2) controls the matching extent from individual node to collective tree. However, it is hard to access the optimal intermediate position, since code editing samples suffer from a huge variance. Therefore, a single extent may not be robust enough to help the model generalize in diverse editing scenarios. We hereby propose a multi-extent measurement to complement the above weakness by setting $\lambda$ with different continuous values, and ensemble all the results $\phi_\theta(t^q_{\lambda_i}, t^{s_k}_{\lambda_i})$ for $\{\lambda_i\}_{i=1}^{N_\lambda}$ to enhance the generalization on complicated code snippets. However, setting $\lambda$ arbitrarily and ensemble all perspectives linearly without selection may corrupt the overall tree representation, and thus deteriorate the model. For a robust ensemble and aggregation, we consider assessing the quality of $\phi_\theta(t^q_{\lambda_i}, t^s_{\lambda_i})$ towards different $\lambda_i$ by using a marginal ranking error on edit representation. Since we have the accessible ground-truth $C^q_+$ during training, we employ the ranking error $e_{\lambda_i}$ to explicitly reflect how good $\phi_\theta(t^q_\lambda, t^{s_k}_\lambda)$ is under a certain $\lambda$ with respect to the similarity between query and $K$ support exemplars. In the inference phase, due to the missing of $C^q_+$, we employ a similarity-ranking error estimator $\hat{e}_{\lambda_i} = R_\mu(t^q_{\lambda_i}, t^{s_1}_{\lambda_i}, t^{s_2}_{\lambda_i}, \ldots, t^{s_K}_{\lambda_i}; \lambda_i) : \mathbb{R}^{(K+1) \times D} \to \mathbb{R}$ to predict the quality of each $\lambda_i$, where $\mu$ is the learnable parameter in $R_\mu$, and ensemble the outcomes. Training details of $R_\mu$ will be elaborated in the next subsection.

To achieve multi-extent composition and ensemble, we first measure the similarity between query and support edit representations via the editing encoder:

$$w_k := \frac{\langle f_\Delta(C^q_-, C^q_+), f_\Delta(C^{s_k}_-, C^{s_k}_+)\rangle}{\left\|f_\Delta(C^q_-, C^q_+)\right\| \cdot \left\|f_\Delta(C^{s_k}_-, C^{s_k}_+)\right\|}. \tag{5}$$

Then we access the error for each $\lambda$ by:

$$e_\lambda := \sum_{k=1}^K \left(w_{\Gamma(1)} - w_{\Gamma(k)}\right) \cdot \max\left\{0, \gamma - \left(\phi_\theta(t^q_\lambda, t^{s_{\Gamma(1)}}_\lambda) - \phi_\theta(t^q_\lambda, t^{s_{\Gamma(k)}}_\lambda)\right)\right\}, \tag{6}$$

where $\Gamma(\cdot)$ is the index mapping for $K$ support exemplars that satisfy $w_{\Gamma(1)} \geq w_{\Gamma(2)} \geq \ldots \geq w_{\Gamma(k)}$, i.e., the mapping to sort $s_k$ in a descending order, and $\gamma$ is the margin set as a hyperparameter. The formulation indicates how well the nearest support exemplars can outperform other support exemplars, and the factor $(w_{\Gamma(1)} - w_{\Gamma(k)})$ reveals the error confidence. Note that the above error term depends on the edit model $f_\Delta$, which is only an approximate estimation. Then we extend Eq. (4) by involving multiple extents and the reciprocal of its error for an adaptive ensemble:

$$f^q_\Delta := \sum_{k=1}^K \sum_{i=1}^{N_\lambda} \frac{1}{e_{\lambda_i} + \epsilon} \phi_\theta(t^q_{\lambda_i}, t^{s_k}_{\lambda_i}) f_\Delta(C^{s_k}_-, C^{s_k}_+). \tag{7}$$

where $\epsilon$ is a constant in denominator for numerical stability. We omit the normalization over $\phi_\theta(t^q_{\lambda_i}, t^{s_k}_{\lambda_i})/(e_{\lambda_i} + \epsilon)$ in the equation. In practice, we apply a softmax function on this term to keep the convexity of the combination. Note that in inference $e_{\lambda_i}$ will be replaced by $\hat{e}_{\lambda_i}$.

## 3.3 LEARNING AND INFERENCE

In the learning phase, we have a set of $K$ support exemplars $\{(C^{s_k}_-, C^{s_k}_+)\}_{k=1}^K$ and one complete query snippet $\{C^q_-, C^q_+\}$ as inputs. Following the Graph2Edit framework (Yao et al., 2021), we obtain the $z^q_n$ and $z^{s_k}_n$ of the $n$-th node representation of the query and $k$-th support snippet and the edit representation $f_\Delta(C^{s_k}_-, C^{s_k}_+)$ for each support exemplar. $L^q$ is the number of steps towards the shortest sequential actions from $C^q_-$ to $C^q_+$, and $a_{1:L^q}$ denotes the corresponding action sequence. Let $\mathbf{e} = (e_{\lambda_1}, e_{\lambda_2}, \ldots, e_{\lambda_N})$ calculated by Eq. (6) and $\hat{\mathbf{e}} = (\hat{e}_{\lambda_1}, \hat{e}_{\lambda_2}, \ldots, \hat{e}_{\lambda_N})$ is its estimation from the estimator $R_\mu$. With $N_\lambda$ extents, our objective function can be written as:

$$\min_{\theta, \mu} -\Pr\left(a_{1:L^q} | f^q_\Delta, t^q\right) + ||\mathbf{e}||_1 + D_{KL}(\hat{\mathbf{e}}\|\mathbf{e}), \tag{8}$$

where $|| \cdot ||_1$ is the vector $l_1$-norm to sum up the absolute values of all elements, $D_{KL}(\cdot \| \cdot)$ is the KL-divergence, $f_\Delta^q$ is the composition of edit representations in Eq. (7), $t^q = \sum_{n=1}^{N_q} z_n^q / N_q$ is the initial tree representation of $C_-^q$, $\theta$ and $\mu$ are the learnable parameters in the similarity measurement $\phi_\theta$, $\varphi_\theta$ and the estimator $R_\mu$. We omit the balanced factor for each term for a simplified expression. The above objective function consists of three parts. The first term maximizes the likelihood between the learned convex combination of edit representations and optimal editing actions, while the rest two terms aim to minimize the ranking error between the query and support snippets at different extent levels and train an estimator to accurately estimate the error for inference usage. The parameters for tree encoder, editing encoder, and decoder are obtained via a pre-training stage on another data split following the self-reconstruction paradigm from Graph2Edit, while their parameters are not tuned in the meta-learning stage for optimization, since we observe a serve overfitting if doing so.

We follow an episode training mechanism (Snell et al., 2017) that is widely used in few-shot learning to simulate the encounter of new editorial pattern in testing. For each training forward, we sample a fixed number of editorial patterns with $K$ support exemplars and one query snippet per class and backpropagate the loss. In the inference phase, given a set of $K$ support exemplars $\{(C_-^{s_k}, C_+^{s_k})\}_{k=1}^K$ and one input query snippet $C_-^q$, we replace $e_{\lambda_i}$ with $\hat{e}_{\lambda_i}$ in Eq. (7) and yields the formulation.

## 4 EXPERIMENT

### 4.1 EXPERIMENTAL SETTING

**Dataset**  We include two code editing datasets for performance evaluation. (1) *C#Fixer* (Yin et al., 2019) is a dataset containing 2,878 editing pairs, generated by applying two C# fixing tools Roslyn[1] and Roslynator[2] on six projects. A fixer is building on top of the C# compiler used to perform refactoring and modernization on codes. One fixer is designed for one type of refactoring and 16 fixers are selected in this dataset. A pre-trained model for *C#Fixer* is trained on *GitHubEdits* dataset using self-reconstruction. The dataset collects 54 C# projects on GitHub which extract the source code before and after the commits from users. We use the data of *GitHubEdits* only for pre-training and follow the training protocol in Yao et al. (2021) but not for few-shot experiments. (2) Benefiting from the flexibility and generalized capacity of abstract syntax description language, beyond C#, we collect another dataset *PyFixer* written in Python 2. We use Python-Future[3], a tool that edits the codes over 5,959 projects written in Python 2 to satisfy the compatibility of Python 3. We obtain the code pairs by taking codes before and after applying the tool and inspecting the changes. To simulate the pre-training and few-shot learning as in *GitHubEdits* and *C#Fixer*, we select 12 types of fixers and extract a fixed number of each fixer from the main data to form the dataset for few-shot learning, while the rest are preserved for pre-training. After data cleaning, we have 14,941 code pairs for (pre) training, 1,664 for validation, and 2,319 for few-shot experiments. The two datasets only contain source code but without any author's name, contact information, or offensive content.

**Baseline**  We employ modifications on two abstract syntax tree-based code editing frameworks, Graph2Tree (Yin et al., 2019) and Graph2Edit (Yao et al., 2021), to fit them into the few exemplars editing setting and involve the following baselines. Note that all baselines are only different in calculating the similarity measure $\phi_\theta$ in Eq. (4). **RS (Random Selection)**: we randomly select one support exemplar from the support set and use its edit representation for standard decoding on query snippet; **NN (Nearest Neighbor)**: we identify the nearest neighbor in the support set based on the distance calculated by graph edit distance (Sanfeliu & Fu, 1983; Abu-Aisheh et al., 2015) over AST trees (**GED-NN**), and cosine similarity over input tree representation (**CS-NN**). **AE (Average Edit Representation)**: we infer the edit representation of $K$ support exemplars using editing encoder respectively, and take the mean of these $K$ representations for query decoding. This is equivalent to setting $\phi_\theta = 1/K$ in Eq. 4. Moreover, we include two other baselines within our proposed composition framework. **GED-Comp (Composition via graph edit distance)** employs the graph edit distance metric as $\phi_\theta$ when comparing the query and the support exemplar among every query-support snippet pair; we take the reciprocal proportion of this metric and form a convex combination over $K$ edit representations. **CS-Comp (Composition via cosine similarity)** means that we employ

---

[1]https://github.com/dotnet/roslyn

[2]https://github.com/josefpihrt/roslynator

[3]https://python-future.org/index.html

Table 1: Experimental results of code editing with 5 support exemplars on *C#Fixer* dataset. RS: random select; AE: average edit representation; GED: graph edit distance; NN: nearest neighbor; CS: cosine similarity; Comp: composition. Please refer to the experiment protocol for a full explanation. The middle line in the table divides the methods into non-composition and composition ones.

| Macro Accuracy | Split #1 | Split #2 | Split #3 | Split #4 | Split #5 | Avg. |
|---|---|---|---|---|---|---|
| Graph2Tree-RS | $0.270 \pm 0.027$ | $0.360 \pm 0.011$ | $0.341 \pm 0.029$ | $0.243 \pm 0.009$ | $0.384 \pm 0.017$ | 0.320 |
| Graph2Edit-RS | $0.279 \pm 0.022$ | $0.423 \pm 0.008$ | $0.413 \pm 0.018$ | $0.225 \pm 0.007$ | $0.396 \pm 0.017$ | 0.347 |
| Graph2Edit-GED-NN | $0.282 \pm 0.022$ | $0.450 \pm 0.005$ | $0.444 \pm 0.018$ | $0.260 \pm 0.011$ | $0.434 \pm 0.036$ | 0.374 |
| Graph2Edit-CS-NN | $0.291 \pm 0.010$ | $0.454 \pm 0.010$ | $0.453 \pm 0.019$ | $0.270 \pm 0.020$ | $0.442 \pm 0.018$ | 0.382 |
| Graph2Tree-AE | $0.381 \pm 0.022$ | $0.363 \pm 0.008$ | $0.372 \pm 0.011$ | $0.275 \pm 0.014$ | $0.349 \pm 0.028$ | 0.348 |
| Graph2Edit-AE | $0.336 \pm 0.025$ | $0.471 \pm 0.012$ | $0.465 \pm 0.022$ | $0.267 \pm 0.018$ | $0.402 \pm 0.023$ | 0.388 |
| Graph2Edit-GED-Comp | $0.363 \pm 0.019$ | $0.479 \pm 0.012$ | $0.487 \pm 0.021$ | $0.302 \pm 0.012$ | $0.457 \pm 0.026$ | 0.418 |
| Graph2Edit-CS-Comp | $0.387 \pm 0.016$ | $0.500 \pm 0.004$ | $0.501 \pm 0.016$ | $0.337 \pm 0.008$ | $0.507 \pm 0.009$ | 0.447 |
| Ours | $\mathbf{0.416} \pm 0.015$ | $\mathbf{0.514} \pm 0.021$ | $\mathbf{0.522} \pm 0.015$ | $\mathbf{0.352} \pm 0.018$ | $\mathbf{0.539} \pm 0.023$ | **0.468** |

| Micro Accuracy | Split #1 | Split #2 | Split #3 | Split #4 | Split #5 | Avg. |
|---|---|---|---|---|---|---|
| Graph2Tree-RS | $0.290 \pm 0.024$ | $0.490 \pm 0.008$ | $0.332 \pm 0.032$ | $0.166 \pm 0.008$ | $0.619 \pm 0.010$ | 0.380 |
| Graph2Edit-RS | $0.281 \pm 0.018$ | $0.538 \pm 0.010$ | $0.417 \pm 0.022$ | $0.167 \pm 0.010$ | $0.599 \pm 0.010$ | 0.400 |
| Graph2Edit-GED-NN | $0.282 \pm 0.020$ | $0.559 \pm 0.003$ | $0.450 \pm 0.019$ | $0.183 \pm 0.006$ | $0.616 \pm 0.016$ | 0.418 |
| Graph2Edit-CS-NN | $0.289 \pm 0.010$ | $0.563 \pm 0.010$ | $0.459 \pm 0.021$ | $0.186 \pm 0.010$ | $0.624 \pm 0.006$ | 0.424 |
| Graph2Tree-AE | $0.391 \pm 0.017$ | $0.546 \pm 0.006$ | $0.353 \pm 0.013$ | $0.177 \pm 0.005$ | $\mathbf{0.673} \pm 0.010$ | 0.428 |
| Graph2Edit-AE | $0.343 \pm 0.019$ | $0.603 \pm 0.011$ | $0.460 \pm 0.025$ | $0.184 \pm 0.012$ | $0.590 \pm 0.020$ | 0.436 |
| Graph2Edit-GED-Comp | $0.367 \pm 0.014$ | $0.607 \pm 0.014$ | $0.485 \pm 0.024$ | $0.200 \pm 0.008$ | $0.606 \pm 0.022$ | 0.453 |
| Graph2Edit-CS-Comp | $0.388 \pm 0.010$ | $0.616 \pm 0.008$ | $0.500 \pm 0.018$ | $0.209 \pm 0.003$ | $0.625 \pm 0.015$ | 0.467 |
| Ours | $\mathbf{0.411} \pm 0.011$ | $\mathbf{0.636} \pm 0.030$ | $\mathbf{0.524} \pm 0.017$ | $\mathbf{0.218} \pm 0.011$ | $0.653 \pm 0.039$ | **0.488** |

mean pooling over query and support trees, compute the similarity $\phi_\theta$ by cosine similarity on tree representations for every query-support snippet pair, normalize, and form a convex combination over the entire available edit representations from support set. Finally, **Ours** is the proposed model which learns the similarity measure $\phi_\theta$ as described in Section 3, where we additionally consider an ensemble of multi-extent similarity scores, as shown in Eq. (7). Note that we find Graph2Tree contains numerous C# specific implementations and is currently not applicable for Python, so we do not include this baseline on *PyFixer*. All the baselines are under MIT license for public usage.

**Protocol** We randomly split fixers for few-shot experiments into meta-train and meta-test sets. For *C#Fixer*, the meta-train set contains 12 types of fixers and the meta-test set has the rest 4 fixers. The meta-train set of *PyFixer* contains 8 fixers and the meta-test set takes the rest 4. To perform a comprehensive empirical evaluation, we repeat the random split procedure and obtain 5 different splits denoted from Split #1 to Split #5. For meta-training, to simulate the low-resource scenario and mitigate the imbalance on the number of samples per fixer, we take 10 samples for each fixer and follow a standard episode training strategy. Every time we construct the support set to train on one query sample, we randomly select the exemplars who are sharing the same fixing type with the query sample in the sampled data. For meta-testing, we iterate through all samples and each of them as a query snippet at a time, then randomly sample from the rest of its fixer peer to build its support set. All numerical results are run with 5 different random seeds. Mean and standard deviation are reported. Note that the variance not only stems from the training per se, but also is affected by the random sampling of support set in testing. Since we fix the random seed every time, all methods are guaranteed to encounter identical query-support combinations in the evaluation under the same seed. All configurable hyperparameters are tuned on a validation split outside the five evaluative splits. We defer the model architecture and configurable hyperparameters to supplementary material. The accuracy per pair is a binary value obtained by whether the model edits $C_-$ to meet exact ground-truth $C_+$. Following Yao et al. (2021), we report both Macro and Micro accuracy since there is a significant fixer imbalance in *C#Fixer*. $K$ is set to 5 and $\lambda$ is set to the ensemble over (1.0, 2.0, 4.0, 10.0) by default if no specification is noted. We instantiate $\varphi_\theta$ (in Eq. (1)) as neural networks and $\phi_\theta$ (in Eq. (4)) as cosine similarity. Experiments are run on a single NVIDIA RTX 3090 graphical card. We complement more technical details in Appendix B.

## 4.2 RESULT AND ANALYSIS

Main numerical results of code editing with 5 support exemplars on *C#Fixer* and *PyFixer* are presented in Table 1 and Table 2. Comparing to baseline methods, we nearly achieve the best results on all splits

Table 2: Experimental results of code editing with 5 support exemplars on *PyFixer* dataset. Macro and micro accuracy are identical on this dataset. The middle line in the table divides the method into non-composition and composition method.

| Macro/Micro Accuracy | Split #1 | Split #2 | Split #3 | Split #4 | Split #5 | Avg. |
|---|---|---|---|---|---|---|
| Graph2Edit-RS | $0.408 \pm 0.017$ | $0.381 \pm 0.015$ | $0.498 \pm 0.033$ | $0.293 \pm 0.007$ | $0.206 \pm 0.006$ | 0.357 |
| Graph2Edit-GED-NN | $0.366 \pm 0.019$ | $0.400 \pm 0.015$ | $0.535 \pm 0.011$ | $0.337 \pm 0.005$ | $0.229 \pm 0.006$ | 0.373 |
| Graph2Edit-CS-NN | $0.446 \pm 0.017$ | $0.425 \pm 0.016$ | $0.541 \pm 0.006$ | $0.346 \pm 0.012$ | $0.239 \pm 0.010$ | 0.399 |
| Graph2Edit-AE | $0.297 \pm 0.012$ | $0.347 \pm 0.008$ | $0.488 \pm 0.010$ | $0.276 \pm 0.008$ | $0.194 \pm 0.009$ | 0.320 |
| Graph2Edit-GED-Comp | $0.337 \pm 0.014$ | $0.368 \pm 0.007$ | $0.509 \pm 0.012$ | $0.297 \pm 0.013$ | $0.219 \pm 0.012$ | 0.346 |
| Graph2Edit-CS-Comp | $0.441 \pm 0.020$ | $0.409 \pm 0.011$ | $0.543 \pm 0.011$ | $0.347 \pm 0.010$ | $0.276 \pm 0.014$ | 0.403 |
| Ours | $\mathbf{0.510} \pm 0.036$ | $\mathbf{0.429} \pm 0.021$ | $\mathbf{0.544} \pm 0.022$ | $\mathbf{0.369} \pm 0.021$ | $\mathbf{0.294} \pm 0.018$ | **0.429** |

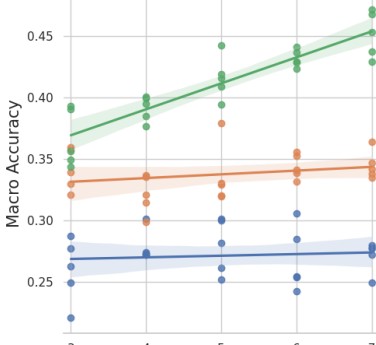 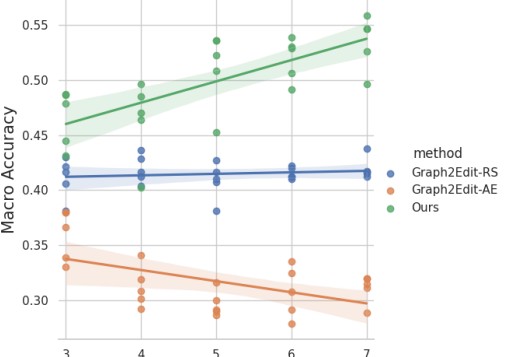

Figure 2: Performance of code editing with various $K$ support exemplars, where the integer $K \in [3, 7]$ along the x-axis. Left is on *C#Fixer* dataset Split #1, and right is on *PyFixer* dataset Split #1. An estimated regression model is plotted for every group of data by the seaborn package, showing as the straight line with deviations. Our approach consistently benefit from adding more support exemplars and is capable to find a better composition, while the other two baselines do not enjoy that.

of two datasets, across macro and micro accuracy. We outperform the second-best baseline method by bringing an absolute improvement range from 8.0% (Macro accuracy on *C#Fixer*) to 10.9% (*PyFixer*). With the same base architecture (exclude Graph2Tree), comparing to GED and CS, our improvements show the significance of multi-extent composition learning of edit representations from support set. Quantitative results demonstrate the effectiveness of treating nodes in abstract syntax tree differently and ensemble learning by the meta-learning strategy. We present extra experiments in this subsection as well as in Appendix C to further investigate our approach. We attach a failure case analysis and a visualization diagram in Appendix D and E, respectively.

**More Support Exemplars**   Figure 2 shows the performance trends when enlarging the size of support set, i.e., offering more resources for code editing. An estimated regression with uncertainty is plotted from $K = 3$ to $K = 7$. It is relatively straightforward that the performance on random selection is not affected by the scale of support set. The average edit representation performs differently on two datasets. Adding more exemplars into support set can help to improve the robustness of the weighted mean representation, or lead it to deviate the representational manifold if high variances exist among exemplars. Our approach consistently benefits from more support exemplars and gradually enlarging the margin towards the two baselines. This demonstrates our model is capable to find a better composition when there are more options, therefore preserving the possibility of capability enhancement when more resources are available.

**Investigation of Multi-Extent Ensemble**   Figure 3 shows the functional analyses on multi-extent measurement and its ensemble, where we investigate seven possible options on extent parameter $\lambda$, four in a single extent, three in a double extent ensemble, and compare to the full model with four levels of extents plotting in red. A single extent in the first four rows underperforms the ensemble when using two or four extents. This confirms our methodological hypothesis that a single extent may not be enough to express the similarity between the query-support snippet pair due to the high variances and that ensembling multiple extents helps the model robustness. The full model with four extents also performs better than the double extent ensemble shown by the median line of the box plot. However, adding more extents implies increasing computational costs and hence a trade-off.

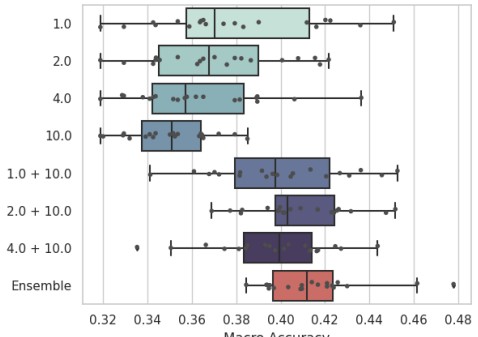 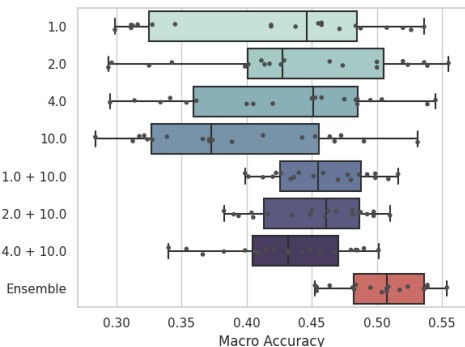

Figure 3: Experiments on multi-extent parameter $\lambda$ and its ensemble. The y-axis represents the instantiated value of $\lambda$, where the plus symbol denotes the ensemble of the two extents, and 'Ensemble' includes all these four. Left: *C#Fixer* dataset Split #1; Right: *PyFixer* dataset Split #1.

## 5 RELATED WORK

**Machine Learning for Code Editing**   Code editing using machine learning has been developed for a long time and is receiving increasing attention in recent years (Menon et al., 2013; Zhang et al., 2019; Tian et al., 2020). Depending on how a code snippet is represented, the research can be roughly summarized into two categories, i.e., performing code editing over code token sequences or abstract syntax tree. In the former category, Yin et al. (2019); Chen et al. (2019); Yasunaga & Liang (2020) directly generate the expected edited code token sequence, while Shin et al. (2018); Vu & Haffari (2018); Dong et al. (2019); Zhao et al. (2019) predict the editing operational sequence, which has been demonstrated to bear more sample efficiency. On the other hand, works in the second category (Yin et al., 2019; Chakraborty et al., 2020; Tarlow et al., 2020; Dinella et al., 2020; Brody et al., 2020; Yao et al., 2021) attempt to edit the trees of programs similarly by either directly generating the edited trees or predicting the tree edit operations. The problem addressed in this paper is conceptually related to 'programming by example' in software engineering that finds the sharing edit operations among different exemplars and applies it to query snippet (Menon et al., 2013; Osera & Zdancewic, 2015; Ferdowsifard et al., 2020; Meng et al., 2013). Recently, Yin et al. (2019); Yao et al. (2021) tackle this problem using deep learning techniques. Nonetheless, both of the two works have focused on code editing from one shot of exemplar, which may not be sufficient for learning generalizable editing, as we elaborated in Section 1. This inspires us to investigate few-shot code editing.

**Few-shot Learning**   Few-shot learning has been developed in many low-resource scenes (Vinyals et al., 2016; Snell et al., 2017; Finn et al., 2017), and with advance in meta seq2seq learning considering the concept of generalized composition (Gu et al., 2018; Lake, 2019; Nye et al., 2020). Our work conceptually relates to the matching principle in few-shot learning (Vinyals et al., 2016; Snell et al., 2017) but with a different compositional task and a specific design for robust multi-extent abstract syntax tree matching. To our best knowledge, we are the first to consider few-shot compositional generalization in code editing, which we believe can inspire development in programming-related applications like code transfer, refactoring, and migration.

## 6 CONCLUSION

In this work, we considered a novel problem in software engineering: code editing with few exemplars. Based on previous frameworks on code editing using abstract syntax tree to present the computer code snippets, we proposed an adaptive multi-extent composition method to perform varying intermediate representations between a collective tree and individual nodes. We leveraged an ensemble approach to gather the query-support matching from multiple extents and delivered a robust composition over support editing representations for the query snippet decoding. Evaluations on two code editing datasets demonstrated the effectiveness of our method over baselines by a large margin. To develop this framework into a product, we consider letting the user collect a few exemplars showing their own purpose. The tool requests an interaction but also offers full flexibility to the user, and is more adapted to a personal development environment.

## 7 ETHICS STATEMENT

We discuss the potential ethical considerations in this section. We do not discover a noteworthy negative societal impact originate from our work. One minor concern is the research in machine learning for source code may encourage and facilitate the code data collection from open-source projects. The data may unintentionally involve user privacy like the author's name, organization, or file date, and the collection process may violate the usage license.

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

## A    EDITING EXAMPLES FOR FIXERS IN *PyFixer*

We list the 12 kinds of fixer names in *PyFixer* in Table 3.

Table 3: Fixer examples in *PyFixer*

| | |
|---|---|
| | Fixer 05: lib2to3.fixes.fix_filter |
| $C_-$ | chunk = filter(lambda x: x.feature == feature, ichunk) |
| $C_+$ | chunk = [x for x in ichunk if x.feature == feature] |
| | Fixer 08: lib2to3.fixes.fix_has_key |
| $C_-$ | key = request.matchdict["key"] if request.matchdict.has_key("key") else "" |
| $C_+$ | key = request.matchdict["key"] if "key" in request.matchdict else "" |
| | Fixer 09: lib2to3.fixes.fix_idioms |
| $C_-$ | return type(self) == type(other) and self.__dict__ == other.__dict__ |
| $C_+$ | return isinstance(self, type(other)) and self.__dict__ == other.__dict__ |
| | Fixer 15: lib2to3.fixes.fix_map |
| $C_-$ | map(int, [a for a in kwargs.values() if a != kwargs['releaselevel']]) |
| $C_+$ | list(map(int, [a for a in kwargs.values() if a != kwargs['releaselevel']])) |
| | Fixer 18: lib2to3.fixes.fix_next |
| $C_-$ | second_gff_chunk = second_gff.next() |
| $C_+$ | second_gff_chunk = next(second_gff) |
| | Fixer 23: lib2to3.fixes.fix_raw_input |
| $C_-$ | raw_input("Start the server on 0:1 and press enter.".format(str(parsed.ip), str(parsed.p))) |
| $C_+$ | input("Start the server on 0:1 and press enter.".format(str(parsed.ip), str(parsed.p))) |
| | Fixer 34: lib2to3.fixes.fix_zip |
| $C_-$ | context.update(zip(inlines_names, kwargs.get('inlines', []))) |
| $C_+$ | context.update(list(zip(inlines_names, kwargs.get('inlines', [])))) |
| | Fixer 34: lib2to3.fixes.fix_absolute_import |
| $C_-$ | from elks import Elks |
| $C_+$ | from .elks import Elks |
| | Fixer 41: lib2to3.fixes.fix_future_standard_library |
| $C_-$ | import urlparse |
| $C_+$ | import urllib.parse |
| | Fixer 42: lib2to3.fixes.fix_future_standard_library_urllib |
| $C_-$ | from urllib2 import HTTPError, urlopen, Request |
| $C_+$ | from urllib.request import urlopen, Request |
| | Fixer 42: lib2to3.fixes.fix_print_with_import |
| $C_-$ | print('site_data_dir', app.locations.site_data_dir) |
| $C_+$ | print(('site_data_dir', app.locations.site_data_dir)) |
| | Fixer 49: lib2to3.fixes.fix_unicode_keep_u |
| $C_-$ | username = unicode(origin.getFrom()).split('/')[1].replace(" ","") |
| $C_+$ | username = str(origin.getFrom()).split('/')[1].replace(" ","") |

## B    EXPERIMENTAL DETAILS

We collect the *PyFixer* dataset by separately applying 51 fixers in Python-Future[4] over 5,959 projects written in Python 2 to satisfy the compatibility of Python 3. We obtain the code pairs by inspecting the changed lines and get the source code before and after the refactoring. We select 12 types of fixers and extract 200 samples for each fixer from the main data to build the dataset for few-shot learning, while the rest are preserved for model pre-training.

---

[4]https://python-future.org/index.html

For training, we set the batch size to 16, epochs to 15, learning rate to 1e-4, with a gradient accumulation for every 2 optimized steps. The extent parameter $\lambda$ for main experiments are set as an ensemble for 10, 5, 2, and 1. The architectural parameters are set the same as Graph2Edit (Yao et al., 2021). The predictor $R$ is implemented with neural networks in two layers with Leaky ReLU. The learnable function $\varphi_\theta(\cdot, \cdot)$ is implemented with one linear layer followed by cosine similarity between the two input terms. The function $\phi(\cdot, \cdot)$ is a cosine similarity function on edit representation. Other detailed implementation or configurations are in our source code since it is not possible to exhasutively include every detail with plain text.

The following equation describes the calculation of macro and micro accuracy. Consider a set of class $\mathcal{C}$, $T_c$ is the number of correctly edited samples in class $c$, $N_c$ is the total number of samples in class $c$. These two metrics can be expressed by

$$\text{Macro accuracy} = \frac{1}{|\mathcal{C}|} \sum_{c \in \mathcal{C}} \frac{T_c}{N_c}, \text{ Micro accuracy} = \frac{\sum_{c \in \mathcal{C}} T_c}{\sum_{c \in \mathcal{C}} N_c}. \tag{9}$$

Note that, in our experiments, an edited code is considered "correct" only when it is exactly the same as the ground truth in both syntax and semantics.

The *PyFixer* are working under Python 3.8, having the abstract syntax tree grammar from https://docs.python.org/3.8/library/ast.html. We list the choice of splits for the two datasets below.

Table 4: Fixer Splits on *C#Fixer*

| | | |
|---|---|---|
| Split #1 | CA2007, IDE0004, RCS1015, RCS1021, RCS1032, RCS1058, RCS1077, RCS1097, RCS1118, RCS1123, RCS1197, RCS1206 | RCS1146, RCS1207, RCS1202, RCS1089 |
| Split #2 | IDE0004, RCS1015, RCS1032, RCS1058, RCS1077, CA2007, RCS1089, RCS1146, RCS1202, RCS1206, RCS1207, RCS1097 | RCS1118, RCS1123, RCS1021, RCS1197 |
| Split #3 | RCS1015, RCS1021, RCS1032, RCS1058, RCS1077, RCS1097, CA2007, IDE0004, RCS1118, RCS1146, RCS1202, RCS1207 | RCS1123, RCS1197, RCS1206, RCS1089 |
| Split #4 | RCS1123, RCS1021, RCS1032, RCS1058, RCS1206, RCS1097, CA2007, IDE0004, RCS1118, RCS1146, RCS1202, RCS1207 | RCS1015, RCS1197, RCS1077, RCS1089 |
| Split #5 | RCS1077, RCS1021, RCS1032, RCS1058, RCS1206, RCS1097, CA2007, IDE0004, RCS1197, RCS1146, RCS1089, RCS1207 | RCS1118, RCS1015, RCS1123, RCS1202 |

Table 5: Fixer Split on *PyFixer*

| | | |
|---|---|---|
| Split #1 | 08, 15, 18, 23, 34, 42, 47, 49 | 05, 41, 09, 35 |
| Split #2 | 08, 09, 35, 23, 34, 42, 47, 05 | 41, 49, 15, 18 |
| Split #3 | 41, 09, 15, 23, 49, 42, 47, 05 | 08, 18, 34, 35 |
| Split #4 | 41, 09, 34, 23, 49, 42, 08, 05 | 47, 15, 18, 35 |
| Split #5 | 15, 09, 34, 23, 49, 35, 08, 47 | 05, 41, 18, 42 |

We use the fixer's name in Table 4 and the fixer's index in Table 5 corresponding to the order in the Python-Future package.

## C    ADDITIONAL EXPERIMENT

We extend our experiments by considering the 'Hit-1-in-5' and 'Hit-5-in-5' cases when editing with one exemplar from the given few exemplars. In the 'Hit-1-in-5' case, we separately decode the query code snippet using the $K$ exemplars and obtain $K$ results. We count the result of the query sample as correct if any of these $K$ decoding results match with the ground truth. Similarly, in the 'Worst' case, we count it as correct when all of these $K$ results are correct. Note that both the 'Hit-1-in-5' and 'Hit-5-in-5' cases are not real in practical usage since we only expect one output from the editing system. Giving possible answers as many as it can is not user-oriented and not scalable. The 'Hit-1-in-5' and 'Hit-5-in-5' cases can be understood as the upper and the lower bound of results if we randomly select one exemplar from the support set but they are almost not achievable. We include an experiment using every exemplar in Figure 4.

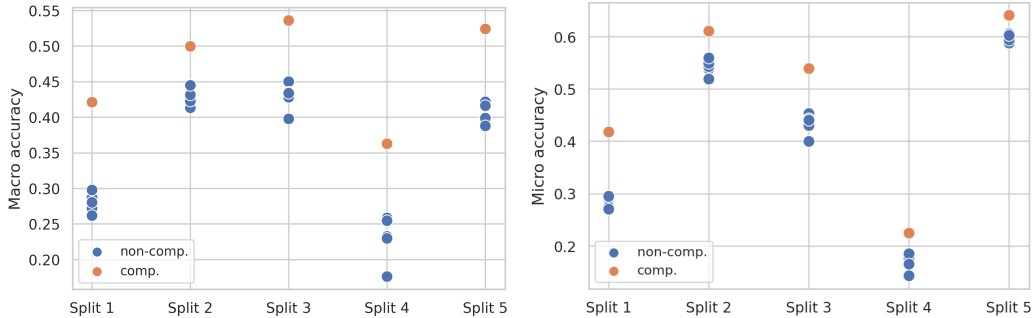

Figure 4: Macro accuracy on *C#Fixer* dataset using Graph2Edit. 'comp.' is abbreviated for composition and 'non-comp.' is abbreviated for non-composition. Blue dots represented the average accuracy over a split obtained by using one exemplar in the support set, indexing from 1 to 5, respectively. Orange dots is the accuracy achieved by our proposed method. Our compositional method consistently outperforms the results using only one exemplar that suffers from variance in code editing.

Table 6: Graph2Edit results of Hit-1-in-5 and Hit-5-in-5 with 5 support exemplars on *C#Fixer* dataset.

| Macro Accuracy | Split #1 | Split #2 | Split #3 | Split #4 | Split #5 | Avg. |
|---|---|---|---|---|---|---|
| Ours | $0.416 \pm 0.015$ | $0.514 \pm 0.021$ | $0.522 \pm 0.015$ | $0.352 \pm 0.018$ | $0.539 \pm 0.023$ | 0.468 |
| Hit-1-in-5 | $0.462 \pm 0.006$ | $0.601 \pm 0.007$ | $0.605 \pm 0.006$ | $0.413 \pm 0.016$ | $0.660 \pm 0.013$ | 0.548 |
| Hit-5-in-5 | $0.072 \pm 0.019$ | $0.154 \pm 0.010$ | $0.126 \pm 0.020$ | $0.045 \pm 0.004$ | $0.111 \pm 0.013$ | 0.102 |

| Micro Accuracy | Split #1 | Split #2 | Split #3 | Split #4 | Split #5 | Avg. |
|---|---|---|---|---|---|---|
| Ours | $0.411 \pm 0.011$ | $0.636 \pm 0.030$ | $0.524 \pm 0.017$ | $0.218 \pm 0.011$ | $0.653 \pm 0.039$ | 0.488 |
| Hit-1-in-5 | $0.443 \pm 0.008$ | $0.721 \pm 0.004$ | $0.616 \pm 0.006$ | $0.312 \pm 0.021$ | $0.780 \pm 0.007$ | 0.574 |
| Hit-5-in-5 | $0.074 \pm 0.017$ | $0.222 \pm 0.012$ | $0.126 \pm 0.022$ | $0.033 \pm 0.003$ | $0.260 \pm 0.017$ | 0.143 |

Hit-1-in-5 requests the ground-truth.

Table 7: Graph2Edit results of Hit-1-in-5 and Hit-5-in-5 with 5 support exemplars on *PyFixer* dataset. Macro and micro accuracy are identical on this dataset.

| Macro/Micro Accuracy | Split #1 | Split #2 | Split #3 | Split #4 | Split #5 | Avg. |
|---|---|---|---|---|---|---|
| Ours | $0.510 \pm 0.036$ | $0.429 \pm 0.021$ | $0.544 \pm 0.022$ | $0.369 \pm 0.021$ | $0.294 \pm 0.018$ | 0.429 |
| Hit-1-in-5 | $0.780 \pm 0.011$ | $0.580 \pm 0.007$ | $0.623 \pm 0.011$ | $0.742 \pm 0.004$ | $0.455 \pm 0.011$ | 0.636 |
| Hit-5-in-5 | $0.059 \pm 0.009$ | $0.107 \pm 0.012$ | $0.165 \pm 0.009$ | $0.045 \pm 0.007$ | $0.013 \pm 0.004$ | 0.078 |

Hit-1-in-5 requests the ground-truth.

Table 8: A failure case from our method on *C#Fixer* dataset

| | Support set |
|---|---|
| $C_-$ | Utils.AssertArgument(this.VAR0!= null && this.VAR0.Any(), LITERAL); |
| $C_+$ | Utils.AssertArgument(this.VAR0?.Any() == true, LITERAL); |
| $C_-$ | bool VAR0= this.VAR1!= null && this.VAR1.Any(); |
| $C_+$ | bool VAR0= this.VAR1?.Any() == true; |
| $C_-$ | var VAR0= VAR1.Properties().Where(VAR2=> VAR2!= null && VAR2.Name != null && String.Equals(VAR2.Name, LITERAL, StringComparison.OrdinalIgnoreCase)); |
| $C_+$ | var VAR0= VAR1.Properties().Where(VAR2=> VAR2?.Name != null && String.Equals(VAR2.Name, LITERAL, StringComparison.OrdinalIgnoreCase)); |
| $C_-$ | var VAR0= VAR1.AsEnumerable().FirstOrDefault(VAR2=> VAR2.Inline != null && VAR2.Inline.Tag == Syntax.InlineTag.String); |
| $C_+$ | var VAR0= VAR1.AsEnumerable().FirstOrDefault(VAR2=> VAR2.Inline?.Tag == Syntax.InlineTag.String); |
| $C_-$ | var VAR0= VAR1.AsEnumerable().FirstOrDefault(VAR2=> VAR2.Inline != null && VAR2.Inline.LiteralContent == LITERAL); |
| $C_+$ | var VAR0= VAR1.AsEnumerable().FirstOrDefault(VAR2=> VAR2.Inline?.LiteralContent == LITERAL); |
| | Query sample |
| $C_-$ | var VAR0= (VAR1!= null) && (VAR1.NodeType == ExpressionType.Not); |
| $C_+$ | var VAR0= (VAR1?.NodeType == ExpressionType.Not); |
| Prediction | var VAR0= (VAR1?.Any() == ExpressionType.Not); |

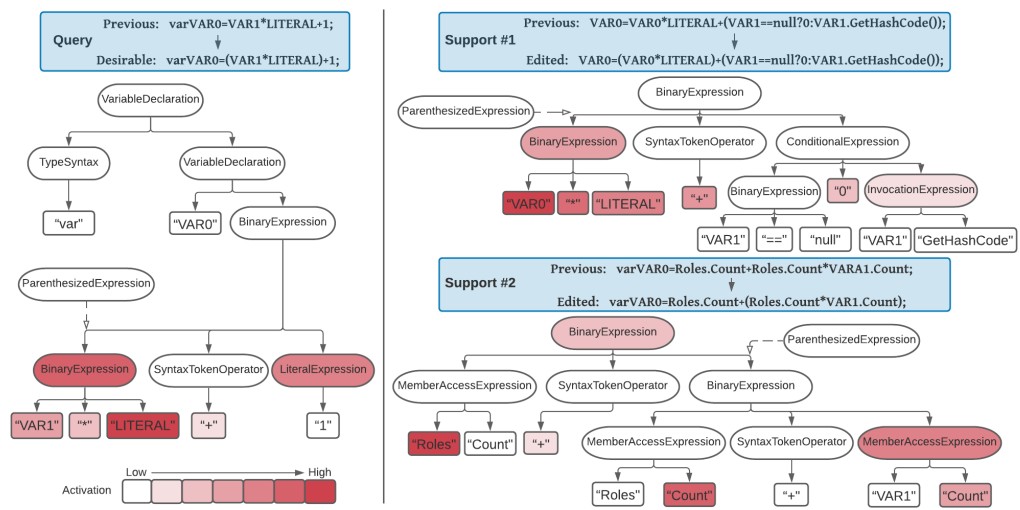

Figure 5: Visualization of normalized node activation with $\lambda = 1.0$ on query snippet and two support exemplars. We plot the top six nodes with the highest activation for each tree and show them in decreasing color saturation. Abstract syntax trees are plotted after pruning on insignificant parts.

## D  FAILURE CASE

We analyze the failure case in this section. A failure case on editing the '&&' logical operation is presented in Table 8. In this case, most of the exemplars showcase the editorial pattern with complicated context, thus it is hard for our algorithm to extract the sharing editing. Meanwhile, two of these exemplars concurrently edit with the attribute 'Any()' and consequently mislead the query code editing.

## E  VISUAL DIAGRAM FOR NODE ACTIVATION ON QUERY-SUPPORT MATCHING

We visualize the node activation in Figure 5. The value of an activation is normalized within the query or support snippet to show the relative magnitude of the value. Matching on individual nodes facilitates query snippet to find the valuable reference in Support #1 under 'BinaryExpression' regardless of the disturbing content under 'ConditionalExpression.' Also, there is a faint activation on the multiple operators in Support #2 since the objects on this operation is under 'MemberAccessExpression.'

