# OpenReview forum: "Code Editing from Few Exemplars by Adaptive Multi-Extent Composition"
_ICLR.cc/2022/Conference — ICLR 2022 Submitted_

### Official Review · Reviewer_L4AR · 2021-10-31

**Correctness:** 3
**Technical Novelty And Significance:** 3
**Empirical Novelty And Significance:** 3
**Recommendation:** 5
**Confidence:** 2

**Main Review:**

This paper addresses an important task in software engineering. Modifying source code towards a desired editing style could be a common programming activity. The paper proposed a new DL-based model that can outperform two existing models.

The results are somewhat unsurprising. It is obvious that providing more examples can improve the generalization ability of a machine learning model. Code editing styles can be better learned from more editing exemplars, so the results are expected.

The related work of this paper is described in a way that is too brief. There has been much work on code similarity and code changes. For example:
Na Meng, et al., LASE: locating and applying systematic edits by learning from examples. In Proc. ICSE  2013, 502–511.

Tian, Haoye, et al. "Evaluating representation learning of code changes for predicting patch correctness in program repair." 2020 35th IEEE/ACM International Conference on Automated Software Engineering (ASE). IEEE, 2020.

J. Zhang, et al., A Novel Neural Source Code Representation based on Abstract Syntax Tree, In Proc. ICSE 2019, May 2019.

Note that the above LASE (Meng et al., 2013) work is a traditional program analysis work that produces systematic edits by learning from multiple examples. The paper can also compare with traditional approaches such as LASE. Currently, it only compared with two deep learning based baseline models.

The most severe issue of this paper is on the evaluation. The evaluation metrics used in the paper, Macro and Micro accuracy,  are not explained. Therefore, it is difficult to interpret the experimental results.

A human evaluation is needed to evaluate the effectiveness of the tool in practice. It would also be better to discuss some unsuccessful examples of code editing. Also, it is not clear how the syntax correctness is assured by the proposed approach.

To capture the multi-extent matching, we design a λ -softmax function by scaling the importance of nodes in an abstract syntax tree. The proposed λ-softmax is derived heuristically . The variable λ is not trainable. So it is difficult to know how it can be learned to reflect the extents of node/tree representations. The presentation can be enhanced if the authors provide more theorical analysis of the proposed method with respect to AST representations.

Minor: In page 7:
bringing a absolute improvement = > bringing an absolute improvement

Our (GDE) => Our (GED)


**Summary Of The Paper:**

This paper presents a few-shot learning approach to editing source code with a few exemplars. Unlike previous work that learns code editing with only one exemplar, the proposed approach learns the edit representations from a few exemplars. The proposed approach combines edit representations extracted from editing exemplars and compositionally generalizes them to the query code snippet editing via multi-extent similarities ensemble. The proposed approach models the extents of node-tree representations using a λ-softmax with an adaptive composition of multiple extents. Experimental results on two code editing datasets demonstrate some improvement over baseline models.

**Summary Of The Review:**

The results are somewhat expected. The evaluation part is unclear. Comparison with more existing work (including traditional approaches) should be performed.

---

> ### Author Response · Authors · 2021-11-16
> **Response to Reviewer L4AR (1/3)**
>
> Thanks for reviewing our paper and bringing these interesting works. We understand your primary concerns are related to relevant works and experiments. We believe there are some misunderstandings so please take time to check our response below.
>
> ---
>
> **1. Clarification on the problem and experimental setting**
>
> Intuitively, extending the number of the exemplar from 1 to few is straightforward and effortless, which we thought at the beginning of this project. However, as the research went deeper, we found the problem was much more difficult than we expected. Actually, we have the same idea with Reviewer ZRFf that the problem should suffice to use a single, most suitable exemplar. We did some exploration along this direction and found that with the ‘complete’ query sample (previous and edited code snippets, where the edited code snippet is the ground truth), it was possible to select the most suitable exemplar from the few candidates and achieve relatively high performance. However, only employing the true query sample (previous code snippet only), we found the nearest neighbor approach on different spaces between support and query snippets struggles to find the best exemplar and leads to unstable performance. Based on the above exploration, we designed the current framework, which learns the support-query matching similarity at multi-extent levels. To make the above point more convincing, we have provided the additional experiments in updated Table 1 and 2.
>
> Moreover, we agree with your opinion that adding more exemplars is good for generalization. We would like to clarify that the baseline approaches (RS and AE), as well as the GED and CS methods, utilize all the provided exemplars instead of only one. The margin between our results and baselines is not due to receiving more exemplars, but the proposed compositional learning strategy on how to utilize more code editing exemplars to enhance generalization.
>
> ---
>
> **2. Related work**
>
> Thanks for bringing these interesting works. In the previous version, we narrowed related works more precisely on code editing with exemplars via machine learning. In the new version, we have added discussions of the three listed works [1-3] in your review to the related work part as well as three ‘programming by example’ papers [4-6] to enrich the scope. We list the added works here for your reference.
>
> [1] Meng, Na, Miryung Kim, and Kathryn S. McKinley. "LASE: locating and applying systematic edits by learning from examples." 2013 35th International Conference on Software Engineering (ICSE). IEEE, 2013.
> [2] Tian, Haoye, et al. "Evaluating representation learning of code changes for predicting patch correctness in program repair." 2020 35th IEEE/ACM International Conference on Automated Software Engineering (ASE). IEEE, 2020.
> [3] Zhang, Jian, et al. "A novel neural source code representation based on abstract syntax tree." 2019 IEEE/ACM 41st International Conference on Software Engineering (ICSE). IEEE, 2019.
> [4] Menon, Aditya, et al. "A machine learning framework for programming by example." International Conference on Machine Learning. PMLR, 2013.
> [5] Osera, Peter-Michael, and Steve Zdancewic. "Type-and-example-directed program synthesis." ACM SIGPLAN Notices 50.6 (2015): 619-630.
> [6]  Ferdowsifard, Kasra, et al. "Small-Step Live Programming by Example." Proceedings of the 33rd Annual ACM Symposium on User Interface Software and Technology. 2020.

---

> > ### Author Response · Authors · 2021-11-16
> > **Response to Reviewer L4AR (2/3)**
> >
> > **3. Compare to LASE**
> >
> > We looked deeply into the LASE paper and its [implementation](https://github.com/Example-based-Program-Transformation/Lase). The traditional LASE method locates and applies the edit with four kinds of actions, and is evaluated on Java programs. LASE identifies the longest subsequence of edit operations from two examples, and applies the edit by context matching. We would like to argue that the method lacks the compositional generalization capacity, and the editing is considered independently of the query sample.
> >
> > Regarding the implementation, their method is working for Java but outside C# and Python. We find no instructions on how to use that in their GitHub repository, and the project has been inactive for years. We contacted the first author of LASE and got a prompt response. However, even with the guidance from the author and tremendous efforts from us, we failed to adapt LASE to our C# and Python datasets. Besides LASE, we have extensively searched the whole GitHub repository for traditional approaches for code editing with examples. We found [Refazer](https://github.com/gustavoasoares/refazer) would be another method for doing the code transformation with exemplars.  However, we still found [issues with the compatibility](https://github.com/gustavoasoares/refazer/issues/36), which are still unsolved by the authors, since the method uses an old version of PROSE framework.
> >
> > We are totally open to comparisons with other baselines, but feel truly exhausted and sincerely sorry for the current infeasible comparison to LASE and other traditional methods. We recognize the reviewer as an expert in both machine learning and software engineering and we would appreciate it if the reviewer could kindly provide a feasible code link. Thank you very much.
> >
> > ---
> >
> > **4. Macro and Micro accuracy**
> >
> > Macro and micro accuracy are used for the evaluation of imbalanced multi-class classification. We follow the evaluation of Yao et al. (2021) to use these metrics. Technically, in experiments, we compare the output from our model to the ground-truth edited code, and result in a True or False for the match. True means the generated code snippet is exactly the same as the ground-truth edited code over every token, while False is not exactly the same. The accuracy reflects how the prediction matches with the ground-truth edited code, including the code semantics as well as the syntax correctness. Since we have multiple types (fixers) of edits with different numbers, macro accuracy will compute the accuracy independently for each class (fixer) and then take the average, hence treating all classes equally, whereas micro accuracy will aggregate the contributions of all classes to compute the average accuracy. The higher the macro/micro accuracy, the better the performance.
> >
> > The following equation describes the calculation of macro and micro accuracy. Consider a set of class $\mathcal{C}$, $T_c$ is the number of correctly edited samples in class $c$, $N_c$ is the total number of samples in class $c$. These two metrics can be expressed by
> >
> > $
> > \begin{equation}
> >     \text{Macro accuracy} = \frac{1}{|\mathcal{C}|}\sum_{c\in \mathcal{C}}\frac{T_c}{N_c},\  \text{Micro accuracy} = \frac{\sum_{c\in\mathcal{C}} T_c}{\sum_{c\in\mathcal{C}} N_c}.
> > \end{equation}
> > $
> >
> > We have added a comprehensive explanation and mathematical formulation on macro and micro accuracy in Appendix B. Here we invite the reviewer to re-interpret the experimental results, and hope it can address your concern.
> >
> > ---
> >
> > **5. Human evaluation**
> >
> > Since our datasets are generated by tools Roslynator and Python-Future, thus in the current step we have the ground-truth for all the query snippets, and hence we can employ the above macro/micro accuracy for evaluation. Human evaluation is definitely a good try in practical scenarios. But to be honest, we have never done this before, and feel a little confused about the concrete procedure. When the human agent evaluates the performance of the outputs from different algorithms, the agent should have the ground truth in mind or in hand for comparison. To us, it is like an annotation behavior. We can still use the above macro/micro accuracy for evaluation. We are confused about the difference between our current evaluation procedure and human evaluation. Are you expecting users’ feedback or something beyond accuracy, such as friendliness and ease of use? Please advise and we are happy to have further discussion.
> >
> > ---
> >
> > **6. Unsuccessful example**
> >
> > We have added analysis on the failure case in **Appendix D and Table 8**. When exemplars are showing the editing in complex and distinct coding content, it is more likely to result in a false prediction.

---

> > > ### Author Response · Authors · 2021-11-16
> > > **Response to Reviewer L4AR (3/3)**
> > >
> > > **7. Syntax correctness**
> > >
> > > While this is not the focus of our work, we would like to clarify that the syntax correctness of each code edit is guaranteed by the Graph2Edit model (Yao et al., 2021). Specifically, Graph2Edit represents each program using the Abstract Syntax Description Language (ASDL) and enforces that the tree edit action at each decoding step is grammatically valid under ASDL. For example, for “optional” fields which allow at most one child, acceptable edit actions include either removing an existing child or adding a new child to an empty field, but not others. In addition, the derivations (e.g., applicable production rules) of each tree node are also restricted under ASDL. We refer the reviewer and other interested readers to Yao et al. (2021, Section 3.1.1) for details.
> > >
> > > ---
> > >
> > > **8. The choice of $\lambda$**
> > >
> > > In our approach, $\lambda$ is the hyperparameter and needs to be heuristically set. To deal with the setting of $\lambda$, we propose a multi-extent composition and ensemble approach in Section 3.2 to make the choice of $\lambda$ robust to the variety of code editing, and consequently, make the setting easier to be done.  A direct result from ensemble learning can be found in Figure 3.
> > >
> > > ---
> > >
> > > **9. Minor**
> > >
> > > We have fixed these typos. Thanks for pointing that out.
> > >
> > > ---
> > >
> > > We appreciate your broad view that helps us improve our paper. We are totally open to any further discussion and more comments are highly welcome!

---

### Official Review · Reviewer_gFho · 2021-11-02

**Correctness:** 4
**Technical Novelty And Significance:** 3
**Empirical Novelty And Significance:** 2
**Recommendation:** 6
**Confidence:** 4

**Main Review:**

The motivation for the problem was well presented, and made sense. A single example edit can lead a model astray, and figuring out which of multiple exemplars to imitate seems like a very useful facility in your approach.

Your query-support matching functions (Section 3.1) have similarities to attention, but some differences as well. For instance, attention is all about alignment, and your similarity functions are essentially trying to align the query graph nodes to the support snippet graph nodes. It would be instructive in the paper to explain the differences and why this is not attention. In general, the formulation of Section 3.1 is not very well motivated and could use some explanation and intuition (e.g., the difference between $m^{q,k_k}$ and $m^{s_k, q}$, and what each captures that the other cannot).

Also, it would be good to explain how you exactly implement the two phi functions in Section 3.1. I had to guess that they are some feedforward network from what you say, but it's only a guess.

I found the term `extent` highly confusing, especially since it's associated with the $\lambda$ parameter, which doesn't have a physical interpretation. I wonder if `resolution` might be a better term, or something else? `Extent` has to do with coverage, or area. If you can intuitively demonstrate how different values of $\lambda$ represent some notion of extent, I'd love to see it.

Something I didn't quite get from your presentation is what makes a good set of exemplars. You argue that more are better for your approach, which makes intuitive sense, but presumably, it's just as easy to have a bunch of exemplars that happen to modify a declaration as in Support #1 of Case a in your example, as it is to have one of those. Your model would be just as likely to be convinced that the rewrite only applies to declaration statements. Perhaps you were lucky in your random draws from exemplars, in that you ended up with a diverse set that improved the prediction. But why would that be the case in general? In fact, what is the use case for this multi-exemplar edit imitation? Are you expecting a developer to write a few example edits and let the model apply them to diverse "queries"? I couldn't quite tell what kind of use case you're anticipating. Will there be noise in the exemplars (i.e., are they always correct?) In your datasets, the exemplars are generated by "oracle" fixers, but I'm guessing in practice you wouldn't have those, or else you wouldn't need your tool in the first place.

Nevertheless, I found the contribution interesting and different from prior related work.

# Questions

Q1: How many exemplars are enough? Is more always better? Is there some measurable metric of fitness for a set of exemplars, e.g., based on diversity or some other similar metric?

Q2: What's the use-case for this multi-exemplar task? Would you expect an editor to provide this functionality? Do you anticipate a different kind of usage mode for this sort of multi-extent input?

Q3: What's the relationship between your query-support matching formulation and standard notions of attention?


# Smaller Issues

1. In equation 1, activations have $g, s_k$ as a superscript. In equation 2, they have $gs_k$. Please be consistent.

2. The descriptions of the "Ours" baselines are confusing. Given the complex formulation of your model, it would help if you pointed out which function of which equation you're replacing with the Graph Edit Distance and cosine similarity, respectively, or just gave a precise alternate formulation.





**Summary Of The Paper:**

This paper tackles the task of applying an edit to code that is similar to some prior edits (think of a linter edit that replaces double quotes with single quotes in Python). Prior work has addressed such a task, by embedding an edit exemplar (before and after snapshots of a file for such an edit), and the before snapshot for a new example, seeking to predict the after snapshot that adapts the edit exemplar to the new before snapshot. However, this work recognizes that generalizing from a single exemplar can push baseline models to overfit the patern modifications of the exemplar.

Instead, this work (a) enables a model to see multiple exemplars, (b) and uses a similarity ranking estimator to weigh different exemplars in a way specific to the "query" (the new before snapshot), so as to influence the prediction accordingly. This similarity formulation is parametrized by how much individual good alignment matches (between exemplars and the new "before" snapshot) should dominate the overall prediction, versus a broadly good alignment match across multiple parts of the various exemplars. An ensemble approach combines multiple values for this parameter.

On a number of datasets generated by applying stock C# and Python fixers to GitHub code, this new approach outperforms baselines.



**Summary Of The Review:**

Task targeted is not well motivated (not clear how you apply it if you don't have the fixers, or why you would use it if you have the fixers). But the approach seems novel and interesting.

---

> ### Author Response · Authors · 2021-11-16
> **Response to Reviewer gFho (1/2)**
>
> Thanks for your review. We greatly appreciate you for recognizing our novel contributions. We understand you have some questions on technical illustrations and the practical usage of our framework at scale. In the following, we address all of them. Please take time to read the following responses for your comments.
>
> ---
>
> **1. The relation to attention mechanism**
>
> At a high level, our proposed query-support matching in Section 3.1 is an attention mechanism. The basic idea is to identify the potentially instructive exemplars and compose their edit representations for editing generalization. Beyond a standard attention mechanism, in order to obtain a robust composition, we conduct the matching at various levels over AST trees from individual nodes to collective tree representation due to the inherent property of code editing. The technique in Section 3 is to compute the matching score of every node in support or query trees towards a given query or support tree. Then, we can adaptively define the scale of coverage of nodes we want to consider in the query-support matching by $\lambda$ and the following method elaborated in Section 3.2 and 3.3. The method in Section 3.1 is conceptually related to the attention mechanism, but involves more specific designs for matching on source code trees.
>
> Regarding $m^{q, s_k}$ and $m^{s_k, q}$, $m^{q, s_k}$ is the matching activation on query sample $q$ incurred by the support sample $s_k$, and similarly, $m^{s_k, q}$ is the matching activation on support sample $s_k$ incur by the query sample q. These two scores are not identical since we use different mapping functions, thus the notations $q$ and $s_k$ are not commutable.
>
> We follow your suggestions and have added these two illustrations in our updated version.
>
> ---
>
> **2. Implementation of $\phi$ and $\varphi$ function**
>
> The $\varphi$ functions in Eq. (1) are implemented with one learnable linear layer followed by the calculation of cosine similarity on the two input terms. The $\phi$ function in Eq. (4) is a cosine similarity function on edit representation. We have updated the technical details in Appendix B ‘Experimental Details’ and pointed readers to these details in the main content.
>
> ---
>
> **3. Term "extent" and its interpretation**
>
> Thanks for providing your understanding of "extent." Indeed, we use the term "extent" to describe the coverage or area of nodes used in query-support matching. A physical interpretation of $\lambda$ has been presented in the paragraph under Eq. (3) in our first draft, while we rephrase the interpretation to make it clear: a larger $\lambda$ denotes greater domination of matched individual nodes in the final tree representation, i.e., the sharpness of the outcomes after $\lambda$-softmax normalization. $\lambda\rightarrow 0$ preserves the final query/support representation as to their initial tree representation (large coverage for support-query matching), and more specifically, $\lambda\rightarrow 0$ calculates a more smoothly weighted average of node representations over the entire tree, which thus implies a larger coverage. $\lambda\rightarrow\infty$ represents the final query/support representation approximately with only a single node representation at most of the time (small coverage for support-query matching), where the node is selected upon the maximum activation from Eq. (1). It tends to approximate the tree representation using the representation of a single tree node (i.e., the tree node which has the maximum matching degree from Eq. (1)), which thus implies a smaller coverage over the tree. For reference, $\lambda$ can also be understood as the inverse “temperature”, which is commonly used to scale the softmax outputs [1].
>
> [1] Hinton, Geoffrey, Oriol Vinyals, and Jeff Dean. "Distilling the Knowledge in a Neural Network." stat 1050 (2015): 9.
>
> ---
>
> **4. Use case**
>
> From a product perspective, the current design mainly expects a developer to submit a few edits and our model could be used to scale this kind of edit. This design leaves full flexibility to users to define what they want, since there are miscellaneous code edits in real development. Also, the effort is affordable since we only request a few samples. To reduce the user effort, we can also provide some typical exemplars for usage or reference. The tool requires interaction but is more adapted to a personal development environment. There could be noise in exemplars in practice, but our current work starts from the pure exemplars first.
>
> We appreciate you expressing your concern from a product perspective. We have added a discussion derived from your concern in the conclusion part.

---

> > ### Author Response · Authors · 2021-11-16
> > **Response to Reviewer gFho (2/2)**
> >
> > **5. Question**
> >
> > 1. About the size and fitness of exemplars. From our experimental results (Figure 2), more exemplars are better for performance. In the future, one could define a measurable metric (e.g., using a threshold upon the distance between the query and the support exemplar in the representation space) to decide whether the collected support exemplars have been sufficient for making unambiguous tree edits. This could be a direction to further enhance the whole pipeline and thanks for sharing this idea.
> > 2. About the use case of our method: Please see our response “4. Use case.”
> > 3. About the relationship between our query-support matching formulation and the standard attention: Please see our response “1. The relation to attention mechanism.”
> >
> > ---
> >
> > **6. Smaller issue**
> >
> > 1. Thanks for pointing out the missing comma in our notations. We have revised Eq. (2) and added a comma between $q$ and $s_k$.
> > 2. We have revised Section 4.1 to clarify our method and its difference compared with the baselines. Generally, all baselines and our method are only different in the calculation of the similarity measure $\phi_\theta$ in Eq (4). Particularly, the two baselines, Graph2Edit-GED-Comp and Graph2Edit-CS-Comp (revised names for clarity), instantiate the similarity measurement using the graph edit distance over ASTs and the cosine similarity over the tree edit representations, respectively. Please refer to our revised Section 4.1 for more details.
> >
> > ---
> >
> > We really appreciate your time spent reviewing our paper, and hope our response entirely addresses your concerns. All the revised parts mentioned in the above responses are colored in red in the updated manuscript. We welcome further discussions and would be happy to address your comments.

---

### Official Review · Reviewer_ZRFf · 2021-11-02

**Correctness:** 4
**Technical Novelty And Significance:** 2
**Empirical Novelty And Significance:** 2
**Recommendation:** 5
**Confidence:** 4

**Main Review:**

The paper aims at generalizing from multiple example edits rather than applying a single given edit to an input program. This is achieved by aggregating the edit representations by matching the input program with the support set of programs from the example edits. The scores for the edit representations are computed by a series of matching steps over the tree representations of the programs. A hyper-parameter $\lambda$ is used to control the granularity at which the tree fragments are matched. The representations of the input program (query) and a support program are computed by node-level cross-attention. This operation is quadratic in the size of ASTs. I was wondering if the tree structure can be used to reduce this cost.

The paper conducts experiments on datasets from two languages: C# and Python with Graph2Tree and Graph2Edit as the underlying neural editors. The results show that the multi-extent matching does better than all other choices.

The paper aggregates edit representations from multiple examples using a number of complex steps. However, the examples come from a single type of fixer and share the same intent. I am not convinced that the paper demonstrates the need to combine multiple edit representations. In particular, it might suffice to use a single, most-suitable examplar. I would therefore like to see single-example baselines that select the nearest neighbor from the support set i) using a discrete measure like tree edit distance over ASTs with terminals and without terminals, and ii) in the vector space. Note that the single example baselines consider in the experiments (Graph2Tree-RS and Graph2Edit-RS) use random selection and all other baselines consider all the examples together. The number of examples is also small in this case. An iterative baseline that takes each of the examples is also required to demonstrate that the combination of multiple examples is indeed better than any of the examples separately. A qualitative discussion of combining multiple edit representations would also be helpful.


**Summary Of The Paper:**

Neural code editors are a class of neural networks which take vector representations of an edit and an input to produce the output by applying the edit to the input program. The existing approaches take a pair of source code snapshots and embed them into an edit representation. This paper argues that a single pair may not be enough to unambiguously perform the edit and multiple examples can help. The authors present an approach to aggregate edit representations from multiple such pairs. The approach is evaluated on code editing datasets in C# and Python. The aggregation approach gives better results than simpler baselines and ablations of the aggregation method.

**Summary Of The Review:**

The paper presents a technique to combine multiple edit representations for neural code editors. The paper is difficult to understand at times. It makes some technical contribution and provides experimental evidence. However, there is a conceptual gap about the need for combining representations from multiple examples. Though at a high-level it is conceivable that multiple examples could help, the paper does not provide evidence that a systematically chosen single example or an iterative approach which goes over all examples and takes the best output would not be sufficient.

---

> ### Author Response · Authors · 2021-11-16
> **Response to Reviewer ZRFf (1/3)**
>
> Thanks for your time reviewing our paper and we appreciate your sharp view. We totally understand your concern because, at the beginning of this project, we had the same consideration that finding the most suitable exemplar could be sufficient. However, as the research went deeper, we found the nearest neighbor approach given by different spaces and metrics between support and query snippets struggles to find the best exemplar, and therefore leads to worse performance. Based on the above exploration, we designed the current compositional framework, which learns the support-query matching similarity at multi-extent levels. Our composition over few exemplars consistently brought improvement. We add the following experimental results based on your proposal to demonstrate the superiority of composition.
>
> ---
>
> **1. Two single-exemplar baselines**
>
> We design to search the nearest neighbor of the query code snippet in the support set. The search is conducted on AST trees (w. terminals) and the tree representation space, computed with graph edit distance and cosine similarity, respectively. We select one nearest exemplar for decoding. These two baselines are updated in **Table 1 and 2** as ‘Graph2Edit-GED-NN’ and ‘Graph2Edit-CS-NN’ for two datasets (5 exemplars), respectively. We copy all results here for your convenience. For more baseline information, please refer to the 'Baseline' paragraph in Section 4.1.
>
> | Macro Accuracy on C#Fixer | Split #1 | Split # 2 | Split # 3 | Split #4 | Split # 5 | Avg. |
> | --- | --- | --- | --- | --- | --- | --- |
> | Graph2Tree-RS | 0.270 $\scriptsize\pm0.027$ | 0.360 $\scriptsize\pm0.011$ | 0.341 $\scriptsize\pm0.029$ | 0.243 $\scriptsize\pm0.009$ | 0.384 $\scriptsize\pm0.017$ | 0.320 |
> | Graph2Edit-RS | 0.279 $\scriptsize\pm0.022$ | 0.423 $\scriptsize\pm0.008$ | 0.413 $\scriptsize\pm0.018$ | 0.225 $\scriptsize\pm0.007$ | 0.396 $\scriptsize\pm0.017$ | 0.347 |
> | Graph2Edit-GED-NN | 0.282 $\scriptsize\pm0.022$ | 0.450 $\scriptsize\pm0.005$ | 0.444 $\scriptsize\pm0.018$ | 0.260 $\scriptsize\pm0.011$ | 0.434 $\scriptsize\pm0.036$ | 0.374 |
> | Graph2Edit-CS-NN | 0.291 $\scriptsize\pm0.010$ | 0.454 $\scriptsize\pm0.010$ | 0.453 $\scriptsize\pm0.019$ | 0.270 $\scriptsize\pm0.020$ | 0.442 $\scriptsize\pm0.018$ | 0.382 |
> | Graph2Tree-AE | 0.381 $\scriptsize\pm0.022$ | 0.363 $\scriptsize\pm0.008$ | 0.372 $\scriptsize\pm0.011$ | 0.275 $\scriptsize\pm0.014$ | 0.349 $\scriptsize\pm0.028$ | 0.348 |
> | Graph2Edit-AE | 0.336 $\scriptsize\pm0.025$ | 0.471 $\scriptsize\pm0.012$ | 0.465 $\scriptsize\pm0.022$ | 0.267 $\scriptsize\pm0.018$ | 0.402 $\scriptsize\pm0.023$ | 0.388 |
> | Graph2Edit-GED-Comp | 0.363 $\scriptsize\pm0.019$ | 0.479 $\scriptsize\pm0.012$ | 0.487 $\scriptsize\pm0.021$ | 0.302 $\scriptsize\pm0.012$ | 0.457 $\scriptsize\pm0.026$ | 0.418 |
> | Graph2Edit-CS-Comp | 0.387 $\scriptsize\pm0.016$ | 0.500 $\scriptsize\pm0.004$ | 0.501 $\scriptsize\pm0.016$ | 0.337 $\scriptsize\pm0.008$ | 0.507 $\scriptsize\pm0.009$ | 0.447 |
> | Ours | **0.416** $\scriptsize\pm0.015$ | **0.514** $\scriptsize\pm0.021$ | **0.522** $\scriptsize\pm0.015$ | **0.352** $\scriptsize\pm0.018$ | **0.539** $\scriptsize\pm0.023$ | **0.468** |
>
> | Micro Accuracy on C#Fixer | Split #1 | Split # 2 | Split # 3 | Split #4 | Split # 5 | Avg. |
> | --- | --- | --- | --- | --- | --- | --- |
> | Graph2Tree-RS | 0.290 $\scriptsize\pm0.024$ | 0.490 $\scriptsize\pm0.008$ | 0.332 $\scriptsize\pm0.032$ | 0.166 $\scriptsize\pm0.008$ | 0.619 $\scriptsize\pm0.010$ | 0.380 |
> | Graph2Edit-RS | 0.281 $\scriptsize\pm0.018$ | 0.538 $\scriptsize\pm0.010$ | 0.417 $\scriptsize\pm0.022$ | 0.167 $\scriptsize\pm0.010$ | 0.599 $\scriptsize\pm0.010$ | 0.400 |
> | Graph2Edit-GED-NN | 0.282 $\scriptsize\pm0.020$ | 0.559 $\scriptsize\pm0.003$ | 0.450 $\scriptsize\pm0.019$ | 0.183 $\scriptsize\pm0.006$ | 0.616 $\scriptsize\pm0.016$ | 0.418 |
> | Graph2Edit-CS-NN | 0.289 $\scriptsize\pm0.010$ | 0.563 $\scriptsize\pm0.010$ | 0.459 $\scriptsize\pm0.021$ | 0.186 $\scriptsize\pm0.010$ | 0.624 $\scriptsize\pm0.006$ | 0.424 |
> | Graph2Tree-AE | 0.391 $\scriptsize\pm0.017$ | 0.546 $\scriptsize\pm0.006$ | 0.353 $\scriptsize\pm0.013$ | 0.177 $\scriptsize\pm0.005$ | **0.673** $\scriptsize\pm\ 0.010$ | 0.428 |
> | Graph2Edit-AE | 0.343 $\scriptsize\pm0.019$ | 0.603 $\scriptsize\pm0.011$ | 0.460 $\scriptsize\pm0.025$ | 0.184 $\scriptsize\pm0.012$ | 0.590 $\scriptsize\pm0.020$ | 0.436 |
> | Graph2Edit-GED-Comp | 0.367 $\scriptsize\pm0.014$ | 0.607 $\scriptsize\pm0.014$ | 0.485 $\scriptsize\pm0.024$ | 0.200 $\scriptsize\pm0.008$ | 0.606 $\scriptsize\pm0.022$ | 0.453 |
> | Graph2Edit-CS-Comp | 0.388 $\scriptsize\pm0.010$ | 0.616 $\scriptsize\pm0.008$ | 0.500 $\scriptsize\pm0.018$ | 0.209 $\scriptsize\pm0.003$ | 0.625 $\scriptsize\pm0.015$ | 0.467 |
> | Ours | **0.411** $\scriptsize\pm0.011$ | **0.636** $\scriptsize\pm0.030$ | **0.524** $\scriptsize\pm0.017$ | **0.218** $\scriptsize\pm0.011$ | 0.653 $\scriptsize\pm0.039$ | **0.488** |

---

> > ### Author Response · Authors · 2021-11-16
> > **Response to Reviewer ZRFf (2/3)**
> >
> > | Macro/Micro Accuracy on PyFixer | Split #1 | Split # 2 | Split # 3 | Split #4 | Split # 5 | Avg. |
> > | --- | --- | --- | --- | --- | --- | --- |
> > | Graph2Edit-RS | 0.408 $\scriptsize\pm0.017$ | 0.381 $\scriptsize\pm0.015$ | 0.498 $\scriptsize\pm0.033$ | 0.293 $\scriptsize\pm0.007$ | 0.206 $\scriptsize\pm0.006$ | 0.357 |
> > | Graph2Edit-GED-NN | 0.366 $\scriptsize\pm0.019$ | 0.400 $\scriptsize\pm0.015$ | 0.535 $\scriptsize\pm0.011$ | 0.337 $\scriptsize\pm0.005$ | 0.229 $\scriptsize\pm0.006$ | 0.373 |
> > | Graph2Edit-CS-NN | 0.446 $\scriptsize\pm0.017$ | 0.425 $\scriptsize\pm0.016$ | 0.541 $\scriptsize\pm0.006$ | 0.346 $\scriptsize\pm0.012$ | 0.239 $\scriptsize\pm0.010$ | 0.399 |
> > | Graph2Edit-AE | 0.297 $\scriptsize\pm0.012$ | 0.347 $\scriptsize\pm0.008$ | 0.488 $\scriptsize\pm0.010$ | 0.276 $\scriptsize\pm0.008$ | 0.194 $\scriptsize\pm0.009$ | 0.320 |
> > | Graph2Edit-GED-Comp | 0.337 $\scriptsize\pm0.014$ | 0.368 $\scriptsize\pm0.007$ | 0.509 $\scriptsize\pm0.012$ | 0.297 $\scriptsize\pm0.013$ | 0.219 $\scriptsize\pm0.012$ | 0.346 |
> > | Graph2Edit-CS-Comp | 0.441 $\scriptsize\pm0.020$ | 0.409 $\scriptsize\pm0.011$ | 0.543 $\scriptsize\pm0.011$ | 0.347 $\scriptsize\pm0.010$ | 0.276 $\scriptsize\pm0.014$ | 0.403 |
> > | Ours | **0.510** $\scriptsize\pm0.036$ | **0.429** $\scriptsize\pm0.021$ | **0.544** $\scriptsize\pm0.022$ | **0.369** $\scriptsize\pm0.021$ | **0.294** $\scriptsize\pm0.018$ | **0.429** |
> >
> > We understand the suggestion ‘with or without terminals’ means removing nodes on specific variable names to prevent the large variation between trees when computing graph edit distance. In practice, we found overall the non-terminal method achieves similar results since we already unified user-defined variable names as ‘VAR.’ Please don’t hesitate to let us know if we misunderstood your question.
> >
> > | Macro Accuracy on C#Fixer | Split #1 | Split # 2 | Split # 3 | Split #4 | Split # 5 | Avg. |
> > | --- | --- | --- | --- | --- | --- | --- |
> > | w. terminals | 0.282 $\scriptsize\pm0.022$ | 0.450 $\scriptsize\pm0.005$ | 0.444 $\scriptsize\pm0.018$ | 0.260 $\scriptsize\pm0.011$ | 0.434 $\scriptsize\pm0.036$ | 0.374 |
> > | w/o terminals | 0.281 $\scriptsize\pm0.011$ | 0.440 $\scriptsize\pm0.010$ | 0.443 $\scriptsize\pm0.019$ | 0.259 $\scriptsize\pm0.016$ | 0.422 $\scriptsize\pm0.031$ | 0.369 |
> >
> > | Micro Accuracy on C#Fixer | Split #1 | Split # 2 | Split # 3 | Split #4 | Split # 5 | Avg. |
> > | --- | --- | --- | --- | --- | --- | --- |
> > | w. terminals | 0.282 $\scriptsize\pm0.020$ | 0.559 $\scriptsize\pm0.003$ | 0.450 $\scriptsize\pm0.019$ | 0.183 $\scriptsize\pm0.006$ | 0.616 $\scriptsize\pm0.016$ | 0.418 |
> > | w/o terminals | 0.282 $\scriptsize\pm0.009$ | 0.545 $\scriptsize\pm0.008$ | 0.449 $\scriptsize\pm0.021$ | 0.180 $\scriptsize\pm0.012$ | 0.595 $\scriptsize\pm0.014$ | 0.410 |
> >
> > | Macro/Micro Accuracy on PyFixer | Split #1 | Split # 2 | Split # 3 | Split #4 | Split # 5 | Avg. |
> > | --- | --- | --- | --- | --- | --- | --- |
> > | w. terminals | 0.366 $\scriptsize\pm0.019$ | 0.400 $\scriptsize\pm0.015$ | 0.535 $\scriptsize\pm0.011$ | 0.337 $\scriptsize\pm0.005$ | 0.229 $\scriptsize\pm0.006$ | 0.373 |
> > | w/o terminals | 0.463 $\scriptsize\pm0.021$ | 0.338 $\scriptsize\pm0.010$ | 0.396 $\scriptsize\pm0.012$ | 0.529 $\scriptsize\pm0.005$ | 0.229 $\scriptsize\pm0.008$ | 0.391 |
> >
> > These results show that searching the nearest neighbor in the AST tree or edit representation space does not achieve the best performance. Our compositional method consistently outperforms the single-exemplar baselines. The experimental results confirm our basic motivation is correct.
> >
> > ---
> >
> > **2. The "Hit-1-in-5" and "Hit-5-in-5" case**
> >
> > We add experimental results in **Table 6 and 7 in Appendix C** (corresponding experimental details are also elaborated in Appendix C) to show the “upper-bound” performance on using a single exemplar if we have the ground-truth ‘nearest’ exemplar in the support set, and contrastively, show the worst/lower-bound performance. Note that the performance is only achieved in a fictitious scenario (requesting ground-truth). In the "Hit-1-in-5" case, we separately decode the query code snippet using $K$ exemplars and obtain $K$ results ($K=5$ in our experiments). We count the result of the query sample as correct if any of these $K$ decoding results match with the ground truth. This setting is equivalent to having the ground truth support exemplar with regard to the decoding accuracy (hence showing “upper-bound” performance). Contrastively, in the "Hit-5-in-5" case, we count it as correct when all of these $K$ results are correct (hence “lower-bound” performance). We copy the results below.

---

> > > ### Author Response · Authors · 2021-11-16
> > > **Response to Reviewer ZRFf (3/3)**
> > >
> > > | Macro Accuracy on C#Fixer | Split #1 | Split # 2 | Split # 3 | Split #4 | Split # 5 | Avg. |
> > > | --- | --- | --- | --- | --- | --- | --- |
> > > | Ours | 0.416 $\scriptsize\pm0.015$ | 0.514 $\scriptsize\pm0.021$ | 0.522 $\scriptsize\pm0.015$ | 0.352 $\scriptsize\pm0.018$ | 0.539 $\scriptsize\pm0.023$ | 0.468 |
> > > | Hit-1-in-5 | 0.462 $\scriptsize\pm0.006$ | 0.601 $\scriptsize\pm0.007$ | 0.605 $\scriptsize\pm0.006$ | 0.413 $\scriptsize\pm0.016$ | 0.660 $\scriptsize\pm0.013$ | 0.548 |
> > > | Hit-5-in-5 | 0.072 $\scriptsize\pm0.019$ | 0.154 $\scriptsize\pm0.010$ | 0.126 $\scriptsize\pm0.020$ | 0.045 $\scriptsize\pm0.004$ | 0.111 $\scriptsize\pm0.013$ | 0.102 |
> > >
> > > | Micro Accuracy on C#Fixer | Split #1 | Split # 2 | Split # 3 | Split #4 | Split # 5 | Avg. |
> > > | --- | --- | --- | --- | --- | --- | --- |
> > > | Ours | 0.411 $\scriptsize\pm0.011$ | 0.636 $\scriptsize\pm0.030$ | 0.524 $\scriptsize\pm0.017$ | 0.218 $\scriptsize\pm0.011$ | 0.653 $\scriptsize\pm0.039$ | 0.488 |
> > > | Hit-1-in-5 | 0.443 $\scriptsize\pm0.008$ | 0.721 $\scriptsize\pm0.004$ | 0.616 $\scriptsize\pm0.006$ | 0.312 $\scriptsize\pm0.021$ | 0.780 $\scriptsize\pm0.007$ | 0.574 |
> > > | Hit-5-in-5 | 0.074 $\scriptsize\pm0.017$ | 0.222 $\scriptsize\pm0.012$ | 0.126 $\scriptsize\pm0.022$ | 0.033 $\scriptsize\pm0.003$ | 0.260 $\scriptsize\pm0.017$ | 0.143 |
> > >
> > > | Macro/Micro Accuracy on PyFixer | Split #1 | Split # 2 | Split # 3 | Split #4 | Split # 5 | Avg. |
> > > | --- | --- | --- | --- | --- | --- | --- |
> > > | Ours | 0.510 $\scriptsize\pm0.036$ | 0.429 $\scriptsize\pm0.021$ | 0.544 $\scriptsize\pm0.022$ | 0.369 $\scriptsize\pm0.021$ | 0.294 $\scriptsize\pm0.018$ | 0.429 |
> > > | Hit-1-in-5 | 0.780 $\scriptsize\pm0.011$ | 0.580 $\scriptsize\pm0.007$ | 0.623 $\scriptsize\pm0.011$ | 0.742 $\scriptsize\pm0.004$ | 0.455 $\scriptsize\pm0.011$ | 0.636 |
> > > | Hit-5-in-5 | 0.059 $\scriptsize\pm0.009$ | 0.107 $\scriptsize\pm0.012$ | 0.165 $\scriptsize\pm0.009$ | 0.045 $\scriptsize\pm0.007$ | 0.013 $\scriptsize\pm0.004$ | 0.078 |
> > >
> > > The results show that if we have the ground-truth "nearest" neighbor w.r.t. decoding results, selecting one exemplar would be sufficient compared to a composition. However, that should not happen. Experiments in the previous part (“1. Two single-exemplar baselines”) demonstrate the failure to identify the ‘ground-truth exemplar. Also, the "Hit-5-in-5" case shows how large the variance is if we select only one exemplar. We include another experiment below to show if we use all exemplars for inference.
> > >
> > > ---
> > >
> > > **3. Taking every exemplar iteratively**
> > >
> > > We experiment to use every exemplar. We present **Figure 4 in Appendix C**, where we use the fixed exemplar index from 1 to 5 in the support set for decoding. The results are averaged over all the query samples in a split and are separately towards the exemplar index. Thus, we have five result points in a split compared to one result point from our compositional method. Results show that our proposed method consistently works better than taking each of the exemplars from the support set.
> > >
> > > ---
> > >
> > > Please let us know whether your concerns on few exemplars are fully resolved. All the experimental results and discussions are organized in the updated paper. We enjoy having further discussions with you.

---

> > > > ### Comment · Reviewer_ZRFf · 2021-11-19
> > > > **Good to see more experimental results**
> > > >
> > > > Thank you very much for taking the efforts to run more experiments to compare against nearest-neighbor and exemplar-by-exemplar iterative baselines.
> > > >
> > > > Let me summarize my understanding, please correct me if I am missing anything:
> > > > 1. The nearest-neighbor (NN) baselines are doing better that the random-selection baselines. "Ours" is still better than NN.
> > > > 2. The iterative baseline (Hit-1-in-5) is outperforming "Ours" consistently. The difference is particularly pronounced for the PyFixer dataset.
> > > >
> > > > The Hit-1-in-5 results show that decoding with each exemplar separately is better than decoding with all exemplars together (using the proposed method). From a practical point of view, a developer would be happy to review 5 predictions if it helps in getting more correct transformations than getting a single but often wrong prediction.
> > > >
> > > > I did not follow part 3 (taking every exemplar iteratively). Are you assigning some index to each exemplar (arbitrarily) and then comparing the results (averaged by index) against "Ours" in Figure 4?

---

> > > > > ### Author Response · Authors · 2021-11-19
> > > > > **Reply to Reviewer ZRFf (Second Round)**
> > > > >
> > > > > Thanks for checking our complementary results. We appreciate the prompt reply from the reviewer.
> > > > >
> > > > > 1. Regarding the first experiment, your understanding is correct. Selecting the nearest neighbor based on some distance is doing better than random selection, but still worse than our compositional method.
> > > > > 2. We believe the reviewer agrees that ‘Hit-1-in-5’ is NOT a fair comparison to our compositional method from an academic perspective. But true, the numerical results of ‘Hit-1-in-5’ outperform our compositional method.
> > > > >
> > > > > From a practical view, we respectfully disagree with the statement ‘a developer would be happy to review 5 predictions if it helps in getting more correct transformations than getting a single but often wrong prediction.’ This statement seems right at the first glance. However, we invite the reviewer to consider the real scenarios in terms of the following three cases below. Those demonstrate reviewing K predictions from ‘Hit-1-in-K’ (K is the number of exemplars) is infeasible for practical usage.
> > > > >
> > > > > 1. **The number of queries is large**. This will happen when we apply this automatic editing tool at scale, which is the main purpose of our tool. Say, we have 1,000 query samples and the number of exemplar is set to 5. Regarding the ratio of correct edits divided by the times of manual reviews, our compositional method gives 468/1,000 = 46.80% correct transformations; however, the manual picking gives 548/5,000 = 10.96% correct transformations. For 1,000 query samples, the manual picking ‘Hit-1-in-5’ requires 5 x 1,000 = 5,000 times of reviewing the results and picking the correct one. Clearly, under the same number of manual reviews, our algorithm can get more correct transformations (46.80% > 10.96%).
> > > > > 2. **The number of exemplar K is large**. Consider a scenario that we have a team of 10 developers and each of them provides 5 exemplars, and thus K=50. Our method consistently benefits from getting a larger K (Figure 2). However, in the manual picking scenario, the outputs can be overwhelming: 50 results per query. It could be exhausting and tedious for a developer to identify the correct transformation by reviewing the 50 results one by one only for one query sample. The effort is comparable for developers to manually complete the editing on their own.
> > > > > 3. **The number of queries and exemplars are both large**. A more common scenario should combine the above two cases, where the ‘Hit-1-in-K’ is an infeasible solution.
> > > > >
> > > > > ---
> > > > >
> > > > > Regarding part 3 and Figure 4, we use the exemplar at each index from 1 to 5 in the support set, i.e., we iteratively use every exemplar in the decoding phase. Then, we average the results at each index and compare them to our compositional method.
> > > > >
> > > > > ---
> > > > >
> > > > > Please let us know whether the above analyses from the practical view make sense to you. We are open to further discussions.

---

### Official Review · Reviewer_4A6k · 2021-11-02

**Correctness:** 3
**Technical Novelty And Significance:** 3
**Empirical Novelty And Significance:** 2
**Recommendation:** 5
**Confidence:** 3

**Main Review:**

Source code editing is a fundamental software engineering task, which has recently attracted a lot of attention in the ML community.
It is used in various SE applications: code refactoring, and program repair.

The paper introduces the problem of code editing from few examples and formulates the method to use multiple examples in training and inference. While the task of code editing from few examples is new, it is conceptually similar to programming by example (PBE). Authors might find it useful to mention program synthesis by examples in related work.

The paper is relatively difficult to read, and some of the terminology is not explained. Is there a precise definition of "editorial style" of source code? Program repair is arguably not a style change, and may lead to a different code semantics (see introduction, first paragraph). On the other hand, programming style is a well established concept which is different from editing style, which needs to be made more clear in text.

It is not clear from the text how to identify helpful support examples for a given dataset for learning stage. Can you describe a procedure/algorithm to select supporting examples?

Baselines: in code editing, alignment plays a crucial role. Does the AE baseline performs average of aligned representations? If not, please consider adding an average with alignment.

Minor:
“composite them to guide editing” -> “combine them to guide editing” or “compose them to guide editing”

Consider re-writing the abstract to highlight the main contributions and the key methods used.

**Summary Of The Paper:**

The paper introduces a method for code editing from few examples. It allows to automatically generalize code edits from few support examples via adaptive multi-extent composition.

The approach is evaluated on two standard code editing datasets (C# fixer and Py fixer) against the Graph2Edit baselines. Specifically, Graph2Edit model variant applied to a randomly selected support example and an average Graph2Edit representation over all K support examples.

The method leads to a 8-10% improvement over baselines.

**Summary Of The Review:**

The paper introduces a task of code editing from few support examples, and formulates the learning and inference methods to implement the composition method on top of the Graph2Edit method. While the evaluation is sound, and shows an improvement of accuracy as a function of number of support exemplars, one practical challenge in applying this technique would be to identify groups of support examples / identifying the edit intent. Which could lead to problems applying this method at scale or in the product.

---

> ### Author Response · Authors · 2021-11-16
> **Response to Reviewer 4A6k (1/2)**
>
> Thanks for your comments and we appreciate you recognizing the significance of our research problem and the soundness of experiments. We take every comment seriously. We found your major concerns raised from some technical expressions and the review of related works. We are happy to provide the following clarifications.
>
> ---
>
> **1. Programming by examples (PBE) in related work**
>
> Thanks for raising this topic and our work is indeed conceptually related to it. Programming by examples describes the scenario that an end-user provides a machine with examples of a task she wishes to perform, and the machine infers a program to accomplish it. Code editing with few exemplars, the task we addressed here, is within this huge concept but has a more specific target, i.e., editing an existing program by leveraging the example code edit pairs. In our revised draft, we have added references [1-5] to and discussions about PBE in the “Related Work” section.
>
> ---
>
> **2. Clarification on the term ‘editorial style’**
>
> We try our best to explain every technical term precisely but may still bring some confusion to some readers. Sorry for that. But we are happy to address all of them and provide a better readability experience to all the readers. Beyond ‘editorial style’, if the reviewer found other terms difficult to follow, we sincerely solicit the reviewer to point them out.
>
> ‘Editorial style’ in the original manuscript describes the type of change between two code snippets. An ‘editorial style’ applied to a code snippet implies modifying codes following a certain functional purpose and operational rule. For the reader's understanding, we present a list of editorial styles that appear in the dataset PyFixer used in experiments in Appendix A Table 5. It contains, for example, the conversion of the ‘next()’ function from Python2 to Python3, the removal of redundant brackets within the ‘print()’ function, and the replacement of the Python dictionary ‘has_key’ method.
>
> For the programming style, we agree with the reviewer that it is different from the editorial style. Programming style is the guidelines to format programming instructions while editorial style emphasizes the code editing process.
>
> For program repair, if the repair can be done by following one code editing rule, it belongs to what we want to describe. We agree using the word ‘style’ may not be suitable for every case we want to include.
>
> Hence, to make the term more precise and not restricted to the simple ‘style’ change, we change the wording from ‘editorial style’ to ‘editorial pattern’ to describe the type of code editing. We have added an additional illustration on ‘editorial pattern’ in the second paragraph in Section 1 to make it more clear, and also added an explanation in the caption of Fig. 1 under the context of Fig. 1. Please let us know if you feel any confusion on other terms. We are eager to address all of them.
>
> ---
>
> **3. How to identify helpful support examples in the learning stage**
>
> Currently, the problem in our paper does not raise any support exemplar selection or identification issue. The support and query snippets in C# and PyFixer datasets are generated by existing fixing tools and every conducted fixing (a data sample in the datasets) has a fixing type. In the learning stage, every time we construct the support set to train on one query sample, we randomly select the exemplars who are sharing the same fixing type defined by the fixing tool with the query sample from the dataset. We have added an explanation in the protocol paragraph in Section 4.1 to make it clear.
>
> ---
>
> **4. Aligned representation in AE baselines**
>
> If we understand the question correctly, you asked if all the representations are in the same output space and thus enable an average over all representations. If so, the answer is yes. All the edit representations of support exemplars come from the same pre-trained editing encoder and are in the same output space, therefore they are aligned. If we misunderstand the question, please don’t hesitate to let us know. Thanks.
>
> ---
>
> [1] Lieberman, Henry, ed. Your wish is my command: Programming by example. Morgan Kaufmann, 2001.
> [2] Menon, Aditya, et al. "A machine learning framework for programming by example." International Conference on Machine Learning. PMLR, 2013.
> [3] Osera, Peter-Michael, and Steve Zdancewic. "Type-and-example-directed program synthesis." ACM SIGPLAN Notices 50.6 (2015): 619-630.
> [4]  Ferdowsifard, Kasra, et al. "Small-Step Live Programming by Example." Proceedings of the 33rd Annual ACM Symposium on User Interface Software and Technology. 2020.
> [5] Meng, Na, Miryung Kim, and Kathryn S. McKinley. "LASE: locating and applying systematic edits by learning from examples." 2013 35th International Conference on Software Engineering (ICSE). IEEE, 2013.

---

> > ### Author Response · Authors · 2021-11-16
> > **Response to Reviewer 4A6k (2/2)**
> >
> > **5. Practical usage**
> >
> > In practice, while collecting support exemplars demands some effort, we believe it is generally affordable for a programmer, especially when our algorithm requires only “few” (e.g., 5) exemplars. This is a reasonable assumption and aligns well with the existing work of “Programming by Example (PBE)”, where a few IO examples are specified for program synthesis. In our revised draft (the “Conclusion” section), we added a discussion to further clarify this assumption.
> >
> > ---
> >
> > **6. Typo**
> >
> > We have changed the sentence to ‘compose them to guide editing’ in the updated version. Thanks for pointing that out.
> >
> > ---
> >
> > **7. Abstract writing**
> >
> > We rephrase the sentences in the abstract to highlight our contributions. Please check the updated version.
> >
> > ---
> >
> > Thanks again for helping us improve our paper. All the revised parts as mentioned in the above point-to-point response can be found in red in the updated paper. We are open to further interactions and more comments are welcome.

---

### Author Response · Authors · 2021-11-16
**General Response and Paper Update**

We would like to express our gratitude for all the valuable reviews! We take every comment very seriously and have revised our manuscript mainly by

1. Involving more related works on code editing that use or do not use machine learning with the corresponding discussion;
2. Extending the experiments on two datasets to show how it will perform if we only consider one exemplar instead of composition. It demonstrates the superiority of compositional learning in few-shot code editing;
3. Adding more intuitive illustrations for some terminologies and concepts to help reading.

All the revisions are colored in red in the updated manuscript. We are open to further discussion and eager to address any raised concerns. Thank you.

---

### Author Response · Authors · 2021-11-22
**Willing to address concerns**

Dear reviewers,

Thanks again for all the insightful feedback from four reviewers! We take every comment seriously and have posted our point-to-point responses as well as the updated manuscript. We are eager to address any concerns regarding our paper, and are able to reply promptly till the end of the discussion period (Nov. 29). Further interactions are highly welcome!

 -- Authors

---

### Decision · Program_Chairs · 2022-01-20

**Decision:**

Reject

**Comment:**

This paper proposes learning to make stylistic code edits (semantics remains similar) based on information from a few exemplars instead of one. The proposed method first parses the code into abstract syntax trees and then use the multi-extent similarity ensemble. This was compared to a Graph2Edit baseline on C# fixer and pyfixer, which are datasets generated by rule-based transcompilers. The proposed method got around 10% accuracy improvement due to a combination of the method and using more than one examplar.

The reviewers find that any improvement due to more examplars to be expected and suggested that 1) one carefully chosen examplar is enough, and 2) that the need for multiple examplars means more practical difficulties in providing them in an application 3) the targets are all generated by rule-based methods and the benefits may not extend to a realistic case where the edits are not so clear and the reviewers wondered about the application value and the potential need for human evaluations.  The authors argued that it is unexpectedly difficult to expand the base method to multiple examplars and users should be able to provide examplars in an application. The authors further provided additional results that addressed some of the reviewer's concerns but the reviewers did not change their evaluation.

Rejection is recommended based on reviewer consensus.